# Expressive Power of Graph Neural Networks for (Mixed-Integer) Quadratic Programs

Ziang Chen [* 1]   Xiaohan Chen [* 2]   Jialin Liu [* 3]   Xinshang Wang [2]   Wotao Yin [2]

## Abstract

Quadratic programming (QP) is the most widely applied category of problems in nonlinear programming. Many applications require real-time/fast solutions, though not necessarily with high precision. Existing methods either involve matrix decomposition or use the preconditioned conjugate gradient method. For relatively large instances, these methods cannot achieve the real-time requirement unless there is an effective preconditioner. Recently, graph neural networks (GNNs) opened new possibilities for QP. Some promising empirical studies of applying GNNs for QP tasks show that GNNs can capture key characteristics of an optimization instance and provide adaptive guidance accordingly to crucial configurations during the solving process, or directly provide an approximate solution. However, the theoretical understanding of GNNs in this context remains limited. Specifically, it is unclear what GNNs can and cannot achieve for QP tasks in theory. This work addresses this gap in the context of *linearly constrained QP* tasks. In the *continuous* setting, we prove that message-passing GNNs can universally represent fundamental properties of convex quadratic programs, including feasibility, optimal objective values, and optimal solutions. In the more challenging *mixed-integer* setting, while GNNs are not universal approximators, we identify a subclass of QP problems that GNNs can reliably represent.

---

[*]Equal contribution   [1]Department of Mathematics, Massachusetts Institute of Technology, Cambridge, MA, United States [2]Decision Intelligence Lab, Damo Academy, Alibaba US, Bellevue, WA, United States [3]Department of Statistics and Data Science, University of Central Florida, Orlando, FL, United States. Correspondence to: Jialin Liu <jialin.liu@ucf.edu>.

*Proceedings of the $42^{nd}$ International Conference on Machine Learning*, Vancouver, Canada. PMLR 267, 2025. Copyright 2025 by the author(s).

## 1. Introduction

**Quadratic programming (QP)** is an important type of optimization problem with applications (Vogelstein et al., 2015; Markowitz, 1952; Rockafellar, 1987). It aims to minimize a quadratic objective function while satisfying specified constraints. When all the constraints are linear, we call a QP problem a linearly constrained quadratic program (LCQP). When they also involve quadratic inequalities, we call the problem a quadratically constrained quadratic program (QCQP). When some variables are restricted to integers, the problem becomes a mixed-integer QP. This study focuses on LCQP and its mixed-integer variant, MI-LCQP.

In many applications, finding solutions quickly is prioritized over perfect precision. For instance, ride-hailing platforms like Uber or Lyft require quick driver-passenger matching to reduce wait times, even without optimal solutions. Similarly, financial trading algorithms must rapidly adjust portfolios to market changes, prioritizing speed over optimality.

Unfortunately, existing methods for QP often rely on computationally expensive techniques such as matrix decomposition and the preconditioned conjugate gradient method (PCG). For example, LU decomposition typically requires $\mathcal{O}(n^3)$ operations for a $n \times n$ matrix (Golub & Van Loan, 2013), though advanced algorithms can achieve lower complexities. The PCG method requires $\mathcal{O}(n^2)$ operations per iteration, with slow convergence for ill-conditioned matrices (Shewchuk, 1994). These challenges highlight the need for novel approaches to meet real-time application demands.

Machine learning brings new chances to QP. Recent research shows that deep neural networks (DNNs) can significantly improve the efficiency when solving QP. Based on DNNs' role, these studies can be categorized as: **(Type I).** DNNs generate adaptive configurations for QP solvers, tailored to specific QP instances, thereby accelerating the solving process. (Bonami et al., 2018; 2022; Ichnowski et al., 2021; Getzelman & Balaprakash, 2021; Jung et al., 2022; King et al., 2024). This approach requires DNNs to capture in-depth features of QP instances and provide customized guidance to the solver. **(Type II).** DNNs replace or warm-start a QP solver. Here, DNNs take in a QP and directly output an approximate solution. These solutions can be used as final

outputs or as initial guesses to accelerate QP solvers (Nowak et al., 2017; Chen et al., 2018; Karg & Lucia, 2020; Wang et al., 2020a;b; 2021; Qu et al., 2021; Gao et al., 2021; Bertsimas & Stellato, 2022; Liu et al., 2022a; Sambharya et al., 2023; Pei et al., 2023; Tan et al., 2024).

**GNNs.** Among the various types of DNNs, this paper focuses on graph neural networks (GNNs) (Scarselli et al., 2008). By conceptualizing QPs as graphs (Figure 1), GNNs can be applied and efficiently handle these tasks (Nowak et al., 2017; Wang et al., 2020b; 2021; Qu et al., 2021; Gao et al., 2021; Tan et al., 2024; Jung et al., 2022). For instance, Wang et al. (2021) use GNNs to solve Lawler's QAP (Lawler, 1963), while Wang et al. (2019); Yu et al. (2020) apply GNNs to Koopman-Beckmann's QAP (Loiola et al., 2007). They exploit key strengths of GNNs: *adaptability to varying graph sizes*, allowing the same model applied to various QPs, and *permutation invariance*, ensuring consistent outputs regardless of node order.

**Expressive power.** Despite their notable advantages, GNNs face fundamental limitations. As Xu et al. (2019) pointed out, GNNs' expressive power is limited: they are not universal approximators for all graph-based functions. Here, *expressive power*, a core concept in deep learning theory, measures the existence of neural networks under a given structure that can approximate (or represent) a broad class of functions, and *universal approximation* guarantees that a model can approximate any functions within its domain.

The contrast between the successful empirical applications of GNNs and their theoretical limitations reveals a significant gap. However, in practice, GNNs do not need to approximate all possible functions but only specific, meaningful mappings relevant to QPs. This leads to the key question motivating this work: *Can GNNs, despite their limitations for general graph-based mappings, exhibit sufficient expressive power to predict the key properties of QPs?*

To address this question, we focus on two types of functions according to the nature of the application. For Type I applications, we investigate whether GNNs can accurately map a QP to its critical features, focusing on the *feasibility* and *optimal objective value*. For Type II, we explore whether GNNs can map a QP to one of its *optimal solutions*. Therefore, we ask:

> *Can GNNs accurately predict the feasibility, optimal objective value, and an optimal solution of a QP?* (1)

The literature has explored the expressive powers of GNNs on general graph tasks (Xu et al., 2019; Azizian & Lelarge, 2021; Geerts & Reutter, 2022; Zhang et al., 2023; Li & Leskovec, 2022; Sato, 2020). However, significant gaps remain in understanding how these results relate to QP. The most relevant works Chen et al. (2023a;b) investigate the expressive power of GNNs for (mixed-integer) linear pro-

grams (MILPs), but their analysis highly depends on the linear structure and does not cover nonlinear programs like QP. A concurrent study by Wu et al. (2024) investigates GNN representations for quadratically constrained quadratic programs (QCQPs) using a tripartite graph structure. While both studies address convex LCQPs, the methodologies and focus differ: Wu et al. (2024) consider general QCQPs, including quadratic constraints, and use a tripartite graph representation, whereas our work focuses on (MI-)LCQPs with a bipartite graph construction. Our analysis also covers mixed-integer settings and proposes an alternative approach to handling quadratic constraints, yielding partially overlapping but distinct results.

**Contributions.** As several studies have empirically shown that incorporating GNNs can significantly enhance the performance of QP solvers, this paper aims to theoretically analyze the expressive power of GNNs in such tasks, identify potential areas for improvement, and highlight key considerations. Specifically, the contributions of this paper include:

- (GNN for LCQP). In the continuous setting, where all variables are allowed to take fractional numbers, we provide an affirmative answer to question (1) with convexity and present a nonconvex counterexample.

- (GNN for MI-LCQP). In mixed-integer settings, where some variables must be integers, we provide counterexamples showing that GNNs cannot universally solve all tasks, giving a negative answer to question (1).

- Despite this limitation, we identify specific, precisely defined subclasses of MI-LCQP where GNNs succeed. Importantly, we also present criteria for determining whether an MI-LCQP belongs to this subclass, which can be efficiently verified numerically.

## 2. Preliminaries

We focus on linearly constrained quadratic programming (LCQP), which is formulated as follows:

$$\min_{x \in \mathbb{R}^n} \quad \frac{1}{2}x^\top Q x + c^\top x, \quad \text{s.t. } Ax \circ b,\ l \le x \le u, \quad (2)$$

where $Q \in \mathbb{R}^{n \times n}$, $c \in \mathbb{R}^n$, $A \in \mathbb{R}^{m \times n}$, $b \in \mathbb{R}^m$, $l \in (\mathbb{R} \cup \{-\infty\})^n$, $u \in (\mathbb{R} \cup \{+\infty\})^n$, and $\circ \in \{\le, =, \ge\}^m$. In this paper, we always assume that $Q$ is symmetric.

**Basic concepts of LCQPs.** An $x$ satisfying all constraints of (2) is named a *feasible solution*. The set of all feasible solutions, $X =: \{x \in \mathbb{R}^n : Ax \circ b,\ l \le x \le u\}$, is referred to as the *feasible set*. The LCQP is *feasible* if this set is non-empty; otherwise, it is infeasible. The value of $\frac{1}{2}x^\top Q x + c^\top x$ is the *objective value*. Its infimum across $X$ is termed the *optimal objective value*. If this infimum is $-\infty$ (the objective value could indefinitely decrease), the LCQP is named *unbounded*. A feasible and bounded LCQP must yield an optimal solution (Eaves, 1971).

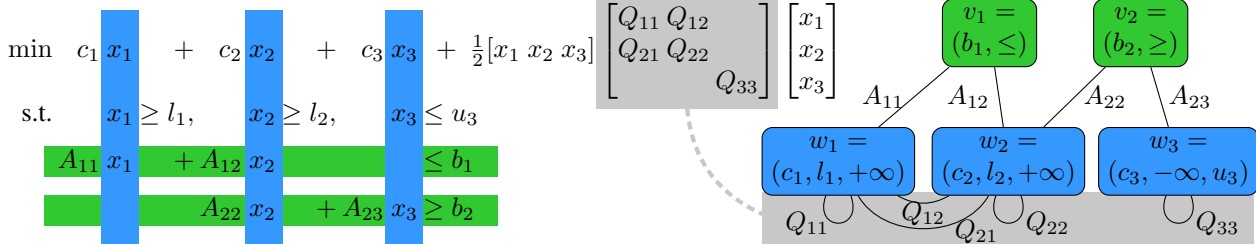

*Figure 1.* LCQP and its graph representation: All the information from the LCQP is fully encoded within the node or edge attributes.

**Graph representation.** We present a graph structure, termed the *LCQP-graph*, $G_{\text{LCQP}} = (V, W, A, Q, H_V, H_W)$, that encodes all the elements of a LCQP. The graph contains two distinct types of nodes: Nodes in $V = \{1, 2, \ldots, m\}$, labeled as $i$, represent the $i$-th constraint and are called *constraint nodes*; Nodes in $W = \{1, 2, \ldots, n\}$, labeled as $j$, represent the $j$-th variable and are known as *variable nodes*. An edge connects $i \in V$ to $j \in W$ if $A_{ij}$ is nonzero, with $A_{ij}$ serving as the edge weight. Similarly, the edge between nodes $j, j' \in W$ exists if $Q_{jj'} \neq 0$, with $Q_{jj'}$ as the edge weight. Self loops ($j = j'$) are permitted. Attributes (or features) $v_i = (b_i, \circ_i)$ are attached to the $i$-th constraint node for $i \in V$. The collection of all such attributes is denoted as $H_V = (v_1, v_2, \ldots, v_m)$. Attributes $w_j = (c_j, \ell_j, u_j)$ are attached to the $j$-th variable node for $j \in W$ and their collection is denoted as $H_W = (w_1, w_2, \ldots, w_n)$.

Such a representation, illustrated by Figure 1, is a fundamental and "minimal" approach in the sense that every entry in $(A, b, c, Q, l, u, \circ)$ is used exactly once. While this particular representation is only detailed in Jung et al. (2022), it forms the foundation of numerous related studies. For instance, removing nodes in $V$ and their associated edges reduces the graph into the assignment graph used in graph matching problems (Nowak et al., 2017; Wang et al., 2020b; 2021; Qu et al., 2021; Gao et al., 2021; Tan et al., 2024). Removing edges associated with $Q$ simplifies the graph to a bipartite structure and reduces LCQP to LP (Chen et al., 2023a; Fan et al., 2023; Liu et al., 2024; Qian et al., 2024). By adding an extra node feature, an approach detailed in Section 4, this graph can also express mixed-integer programs (Gasse et al., 2019; Chen et al., 2023b; Nair et al., 2020; Gupta et al., 2020; Shen et al., 2021; Gupta et al., 2022; Khalil et al., 2022; Paulus et al., 2022; Scavuzzo et al., 2022; Liu et al., 2022b; Huang et al., 2023; Wang et al., 2024).

**GNNs for LCQPs.** Given the graph representation, we present *message-passing graph neural networks (hereafter referred to simply as GNNs)* for LCQPs. They take in an LCQP-graph $G_{\text{LCQP}}$ (including the node and edge attributes) and update node attributes sequentially across layers via a message-passing mechanism. Initially, node attributes are updated separately using embedding mappings $f_0^V, f_0^W$:

$$s_i^0 = f_0^V(v_i) \text{ for } i \in V, \text{ and } t_j^0 = f_0^W(w_j) \text{ for } j \in W.$$

The architecture includes $L$ standard **message-passing** layers where each layer (where $1 \leq l \leq L$) updates node attributes by locally aggregating neighbor information:

$$s_i^l = f_l^V\Big(s_i^{l-1}, \sum_{j \in \mathcal{N}_i^W} g_l^W(t_j^{l-1}, A_{ij})\Big)$$

$$t_j^l = f_l^W\Big(t_j^{l-1}, \sum_{i \in \mathcal{N}_j^V} g_l^V(s_i^{l-1}, A_{ij}), \sum_{j' \in \mathcal{N}_j^W} g_l^Q(t_{j'}^{l-1}, Q_{jj'})\Big)$$

where $f_l^V, f_l^W, g_l^V, g_l^W, g_l^Q$ are trainable local updates in GNNs and $\mathcal{N}_i^W = \{j \in W : A_{ij} \neq 0\}$, $\mathcal{N}_j^V = \{j \in V : A_{ij} \neq 0\}$, and $\mathcal{N}_j^W = \{j' \in W : Q_{jj'} \neq 0\}$ are the sets of neighbors. Finally, there are two types of output layers. For applications where the GNN maps LCQP-graphs to a real value, such as evaluating properties like feasibility of LCQP, a **graph-level output** layer is employed that computes a single real number encompassing the entire graph:

$$y = r_1\Big(\sum_{i \in V} s_i^L, \sum_{j \in W} t_j^L\Big) \in \mathbb{R}.$$

Alternatively, if the GNN is required to map the LCQP-graph to a vector $y \in \mathbb{R}^n$, assigning a real number to each variable node as its output (as is typical in applications where GNNs are used to predict solutions), then a **node-level output** should be utilized:

$$y_j = r_2\Big(\sum_{i \in V} s_i^L, \sum_{j \in W} t_j^L, t_j^L\Big), \quad \text{for } j \in W.$$

Here, $r_1$ and $r_2$ are trainable output functions. In our theoretical analysis, we assume all the mappings $f_l^V, f_l^W$ ($0 \leq l \leq L$), $g_l^V, f_l^W, g_l^Q$ ($1 \leq l \leq L$), and $r_1, r_2$ to be continuous. In practice, these continuous mappings are usually parameterized by multilayer perceptrons (MLPs) and their parameters are learned from data.

**Definition 2.1** (Space of LCQP-graphs). The set of all LCQP-graphs, denoted as $\mathcal{G}_{\text{LCQP}}^{m,n}$ [1], comprises graphs with $m$ constraints and $n$ variables, where $Q$ is symmetric.

**Definition 2.2** (Spaces of GNNs). The collection of all GNNs, denoted as $\mathcal{F}_{\text{LCQP}}$ for graph-level outputs (or $\mathcal{F}_{\text{LCQP}}^W$

---

[1]The space $\mathcal{G}_{\text{LCQP}}^{m,n}$ is equipped with the subspace topology induced from the product space $\{(A, b, c, Q, l, u, \circ) : A \in \mathbb{R}^{m \times n}, b \in \mathbb{R}^m, c \in \mathbb{R}^n, Q \in \mathbb{R}^{n \times n}, l \in (\mathbb{R} \cup \{-\infty\})^n, u \in (\mathbb{R} \cup \{+\infty\})^n, \circ \in \{\leq, =, \geq\}^m\}$. All Euclidean spaces have standard Eudlidean topologies, discrete spaces are equipped with the discrete topology, and their unions are disjoint unions.

for node-level outputs), consists of all GNNs constructed using continuous mappings $f_l^V, f_l^W$ $(0 \leq l \leq L)$, $g_l^V, f_l^W, g_l^Q$ $(1 \leq l \leq L)$, and $r_1$ (or $r_2$).

**Definition 2.3** (Target mappings). We define:

- Feasibility mapping: $\Phi_{\text{feas}}(G_{\text{LCQP}}) = 1$ if $G_{\text{LCQP}}$ is feasible and $\Phi_{\text{feas}}(G_{\text{LCQP}}) = 0$ if it is infeasible.

- Optimal objective mapping: $\Phi_{\text{obj}}(G_{\text{LCQP}}) \in \mathbb{R} \cup \{\pm\infty\}$ computes the optimal objective value of $G_{\text{LCQP}}$. $\Phi_{\text{obj}}(G_{\text{LCQP}}) = +\infty$ means the problem is infeasible and $\Phi_{\text{obj}}(G_{\text{LCQP}}) = -\infty$ means unboundedness.

- Optimal solution mapping: For a feasible and bounded LCQP problem (i.e., $\Phi_{\text{obj}}(G_{\text{LCQP}}) \in \mathbb{R}$), an optimal solution exists though it might not be unique. However, the optimal solution with the smallest $\ell_2$-norm must be unique if $Q \succeq 0$, i.e., $Q$ is positive semi-definite, and we define it as $\Phi_{\text{sol}}(G_{\text{LCQP}})$.

Given the definitions above, we can formally pose the question in (1) as follows: Is there any $F \in \mathcal{F}_{\text{LCQP}}$ that well approximates $\Phi_{\text{feas}}$ or $\Phi_{\text{obj}}$? Similarly, is there any function $F_W \in \mathcal{F}_{\text{LCQP}}^W$ that well approximates $\Phi_{\text{sol}}(G_{\text{LCQP}})$?

# 3. Universal approximation for convex LCQPs

This section presents our main theoretical results for the expressive power of GNNs for representing properties of LCQPs. In particular, we show that for any convex LCQP data distribution, there always is a GNN that can predict LCQP properties, in the sense of universally approximating target mappings in Definition 2.3, within a given error tolerance. Although it is known in the previous literature that there exists some continuous function that cannot be approximated by GNNs with arbitrarily small error, see e.g., Xu et al. (2019); Azizian & Lelarge (2021); Geerts & Reutter (2022), our results in this section indicate that approximating the target mappings of LCQPs do not suffer from this limitation.

**Assumption 3.1.** $\mathbb{P}$ is a Borel regular probability measure defined on the space of LCQP-graphs $\mathcal{G}_{\text{LCQP}}^{m,n}$.

The assumption of Borel regularity is generally satisfied for most data distributions in practice.

**Theorem 3.2.** *For any $\mathbb{P}$ satisfying Assumption 3.1 and any $\epsilon > 0$, there exists $F \in \mathcal{F}_{\text{LCQP}}$ such that $\mathbb{I}_{F(G_{\text{LCQP}})>\frac{1}{2}}$ acts as a classifier for LCQP-feasibility, with an error of up to $\epsilon$:*

$$\mathbb{P}\left[\mathbb{I}_{F(G_{\text{LCQP}})>\frac{1}{2}} \neq \Phi_{\text{feas}}(G_{\text{LCQP}})\right] < \epsilon,$$

*where $\mathbb{I}$. is the indicator function: $\mathbb{I}_{F(G_{\text{LCQP}})>\frac{1}{2}} = 1$ if $F(G_{\text{LCQP}}) > \frac{1}{2}$; $\mathbb{I}_{F(G_{\text{LCQP}})>\frac{1}{2}} = 0$ otherwise.*

This result suggests that a GNN is a universal classifier for LCQP feasibility: for any data distribution of LCQPs satisfying Assumption 3.1, there exists a GNN that can classify LCQP feasibility with arbitrarily high accuracy. This is a natural extension of the feasibility classification for

linear programs (Chen et al., 2023a), as feasibility is solely determined by the constraints, independent of the objective function, and all LCQP constraints are linear.

However, using GNNs to predict the optimal objective value or an optimal solution is highly non-trivial due to the non-linear term $x^\top Q x$. Fortunately, when restricting LCQPs to convex cases, GNNs can universally represent the optimal objective value and an optimal solution for these LCQPs.

**Theorem 3.3.** *For any $\mathbb{P}$ satisfying Assumption 3.1 with $\mathbb{P}[Q \succeq 0] = 1$ ($Q$ is positive semidefinite almost surely), and for any $\epsilon > 0$, there exists $F_1 \in \mathcal{F}_{\text{LCQP}}$ such that*

$$\mathbb{P}\left[\mathbb{I}_{F_1(G_{\text{LCQP}})>\frac{1}{2}} \neq \mathbb{I}_{\Phi_{\text{obj}}(G_{\text{LCQP}})\in\mathbb{R}}\right] < \epsilon. \quad (3)$$

*Addtitionally, if $\mathbb{P}[\Phi_{\text{obj}}(G_{\text{LCQP}}) \in \mathbb{R}] = 1$, then for any $\epsilon, \delta > 0$, there exists $F_2 \in \mathcal{F}_{\text{LCQP}}$ such that*

$$\mathbb{P}\left[|F_2(G_{\text{LCQP}}) - \Phi_{\text{obj}}(G_{\text{LCQP}})| > \delta\right] < \epsilon. \quad (4)$$

This theorem indicates that GNNs can approximate the optimal objective value mapping $\Phi_{\text{obj}}$ very well in two senses: (I) GNN can predict whether the optimal objective value is a real number or $\pm\infty$, i.e., whether the LCQP problem is feasible and bounded or not. (II) For a data distribution over feasible and bounded LCQP problems, GNN can approximate the optimal objective. Finally, we prove that GNN can approximate the optimal solution map $\Phi_{\text{sol}}$.

**Theorem 3.4.** *For any $\mathbb{P}$ satisfying Assumption 3.1 with $\mathbb{P}[Q \succeq 0] = \mathbb{P}[\Phi_{\text{obj}}(G_{\text{LCQP}}) \in \mathbb{R}] = 1$, and for any $\epsilon, \delta > 0$, there exists $F_W \in \mathcal{F}_{\text{LCQP}}^W$ such that*

$$\mathbb{P}\left[\|F_W(G_{\text{LCQP}}) - \Phi_{\text{sol}}(G_{\text{LCQP}})\| > \delta\right] < \epsilon.$$

The detailed proofs of Theorems 3.3 and 3.4 will be presented in Appendix A. We briefly describe the main idea here. The Stone-Weierstrass theorem and its variants are a powerful tool for proving universal-approximation-type results. Recall that the classic version of the Stone-Weierstrass theorem states that under some assumptions, a function class $\mathcal{F}$ can uniformly approximate every continuous function if and only if it **separates points**, i.e., for any $x \neq x'$, one has $F(x) \neq F(x')$ for some $F \in \mathcal{F}$. Otherwise, we say $x$ and $x'$ are **indistinguishable** by any $F \in \mathcal{F}$. Therefore, the key component in the proof is to establish some separation results in the sense that two LCQP-graphs with different optimal objective values (or different optimal solutions with the smallest $\ell_2$-norm) must be distinguished by some GNN in the class $\mathcal{F}_{\text{LCQP}}$ (or $\mathcal{F}_{\text{LCQP}}^W$). It is shown in Xu et al. (2019); Azizian & Lelarge (2021); Geerts & Reutter (2022) that the **separation power**[2] of GNNs is equivalent to the Weisfeiler-Lehman (WL) test (Weisfeiler & Leman, 1968), a classical algorithm for the graph isomorphism problem. We show that, *any two LCQP-graphs that are indistinguishable by*

---

[2] Given two sets of functions, $\mathcal{F}$ and $\mathcal{F}'$, both defined over domain $X$, if $\mathcal{F}$ separating points $x$ and $x'$ implies that $\mathcal{F}'$ also separates $x$ and $x'$ for any $x, x' \in X$, then the separation power of $\mathcal{F}'$ is considered to be stronger than or equal to that of $\mathcal{F}$.

$$\min_{x \in \mathbb{R}^7} \frac{1}{2} x^\top \mathbf{1}\mathbf{1}^\top x + \mathbf{1}^\top x,$$

$$\text{s.t. } x_1 - x_2 = 0, \ x_2 - x_1 = 0,$$
$$x_3 - x_4 = 0, \ x_4 - x_5 = 0, \quad (5)$$
$$x_5 - x_6 = 0, \ x_6 - x_7 = 0, \ x_7 - x_3 = 0,$$
$$x_1 + x_2 + x_3 + x_4 + x_5 + x_6 + x_7 = 6$$
$$0 \le x_j \le 3, \forall \, j \in \{1, 2, \ldots, 7\}.$$

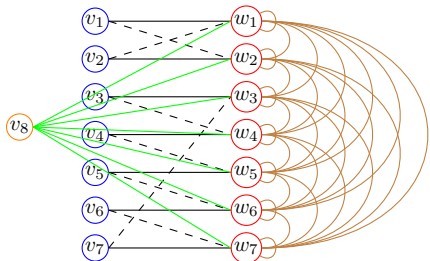

$$\min_{x \in \mathbb{R}^7} \frac{1}{2} x^\top \mathbf{1}\mathbf{1}^\top x + \mathbf{1}^\top x,$$

$$\text{s.t. } x_1 - x_2 = 0, \ x_2 - x_3 = 0, \ x_3 - x_1 = 0,$$
$$x_4 - x_5 = 0, \ x_5 - x_6 = 0, \quad (6)$$
$$x_6 - x_7 = 0, \ x_7 - x_4 = 0,$$
$$x_1 + x_2 + x_3 + x_4 + x_5 + x_6 + x_7 = 6$$
$$0 \le x_j \le 3, \forall \, j \in \{1, 2, \ldots, 7\}.$$

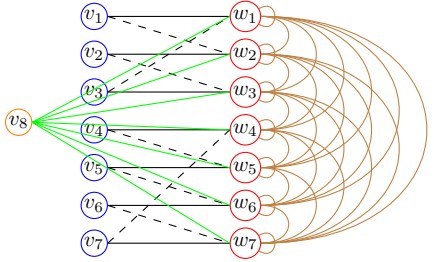

*the WL test, or equivalently by all GNNs, even if they are not isomorphic, they must have identical optimal objective value and identical optimal solution with the smallest $\ell_2$-norm.*

**An illustrative example.** We use the two LCQPs, (5) and (6), each with 7 variables and 8 constraints, to illustrate our findings. All variable nodes share the attribute $w_j = (1, 0, 3)$ for $1 \le j \le 7$, which represents an objective coefficient of $c_j = 1$, lower bound $l_j = 0$, upper bound $u_j = 3$. We refer to these nodes as "red nodes." The first seven constraint nodes $v_i$ (for $1 \le i \le 7$) are assigned the same attribute, $v_i = (0, =)$, which we label as "blue nodes". The eighth constraint node $v_8$ is unique, with the attribute $v_8 = (6, =)$, and is called the "brown node." Any red node is connected to a blue node with weight $A_{ij} = 1$ (solid lines), another blue node with weight $A_{ij} = -1$ (dashed lines), the brown node with weight $A_{ij} = 1$ (green lines), and all seven red nodes with $Q_{jj'} = 1$ (brown curves).

*(Observation I).* The two LCQPs are not equivalent, and their graph representations are not isomorphic. Both LCQPs are feasible and bounded, with identical optimal objective values, as $\mathbf{1}^\top x = 6$ results in $\frac{1}{2} x^\top \mathbf{1}\mathbf{1}^\top x + \mathbf{1}^\top x = 24$. However, they are not equivalent because their solution sets differ. Specifically, $(3, 3, 0, 0, 0, 0, 0)$ is a solution to (5) but not to (6), while $(2, 2, 2, 0, 0, 0, 0)$ is a solution to (6) but not to (5). Furthermore, their graph representations differ: in the first graph, the 7 variables form two groups—one with 2 nodes and the other with 5—where nodes in each group are connected cyclically. In the second graph, the variables form two groups, but one with 3 nodes and the other with 4.

*(Observation II).* The two graphs cannot be distinguished by any GNNs, even after multiple rounds of message passing. Initially, both graphs are indistinguishable as they share identical node attributes: seven red nodes, seven blue nodes, and one brown node. During message passing, each red node receives identical updates due to symmetric connections: one blue node with weight $A_{ij} = 1$, another with $A_{ij} = -1$, the brown node with $A_{ij} = 1$, and all seven red nodes with $Q_{jj'} = 1$. As a result, the red node's attribute is updated as (an informal but illustrative equation):

$$t_j^l = f_l^w \Big( \text{red node}, g_l^V(\text{blue node}, 1) + g_l^V(\text{blue node}, -1)$$
$$+ g_l^V(\text{brown node}, 1), \ 7 \cdot g_l^Q(\text{red node}, 1) \Big).$$

After the update, all red nodes $t_j^l (1 \le j \le 7)$ in both graphs retain identical attributes and are still indistinguishable. The same applies to the blue and brown nodes. Therefore, regardless of how many message-passing rounds occur, both graphs will still have seven red nodes, seven blue nodes, and one brown node. This holds for any parameterized mappings used in GNNs ($f_l^V, f_l^W, g_l^V, g_l^W,$ and $g_l^Q$), meaning no GNN can differentiate between the two instances.

*(Observation III).* They share the common optimal solution with the smallest $\ell_2$-norm: $(\frac{6}{7}, \frac{6}{7}, \frac{6}{7}, \frac{6}{7}, \frac{6}{7}, \frac{6}{7}, \frac{6}{7})$, which follows from $\mathbf{1}^\top x = 6$ and the Cauchy-Schwarz inequality.

Note that these observations are not mere coincidences; we have established this conclusion in general for LCQPs (see Definition A.1, Theorem A.2, and Theorem A.3).

Additionally, similar results can be extended to QCQPs under certain assumptions, though it is beyond the main focus of this paper. The details are deferred to Appendix E.

**A nonconvex counterexample.** At the end of this section, we comment that Theorems 3.3 and 3.4 do not hold if the convexity assumption ($\mathbb{P}[Q \succeq 0] = 1$) is removed. In particular, consider a convex LCQP

$$\min \ \frac{1}{2} \begin{pmatrix} x_1 & x_2 \end{pmatrix} \begin{pmatrix} 1 & 0 \\ 0 & 1 \end{pmatrix} \begin{pmatrix} x_1 \\ x_2 \end{pmatrix}, \ \text{s.t.} -1 \le x_1, x_2 \le 1,$$

and a nonconvex LCQP

$$\min \ \frac{1}{2} \begin{pmatrix} x_1 & x_2 \end{pmatrix} \begin{pmatrix} 0 & 1 \\ 1 & 0 \end{pmatrix} \begin{pmatrix} x_1 \\ x_2 \end{pmatrix}, \ \text{s.t.} -1 \le x_1, x_2 \le 1.$$

Following the aforementioned argument, one can verify that these two LCQP instances are indistinguishable by any GNNs. On the other hand, the first convex LCQP has an optimal objective being $0$ and the optimal solution being $(0, 0)$, while the second nonconvex LCQP has an optimal objective being $-1$ and optimal solutions being $(1, -1)$ and $(-1, 1)$. This indicates that the conclusions in Theorems 3.3 and 3.4 do not hold true for the data distribution that is supported on aforementioned instances with equal probability, since a GNN making accurate prediction on one instance must fail on the other.

## 4. The capacity of GNNs for MI-LCQPs

In this section, we turn to mixed-integer linearly constrained quadratic program (MI-LCQP), which is almost the same as LCQP (2) except that some entries of $x$ are constrained to be integers: $x_j \in \mathbb{Z}$, $\forall j \in I$, where $I \subset \{1, 2, \ldots, n\}$ collects the indices of all integer variables.

**MI-LCQP-graph** is modified from the LCQP-graph by adding a new entry to the feature of each variable node $j \in W$. The new feature is $w_j = (c_j, l_j, u_j, \delta_I(j))$ where $\delta_I(j) = 1$ if $j \in I$ and $\delta_I(j) = 0$ otherwise. We use $\mathcal{G}_{\text{MI-LCQP}}^{m,n}$ to denote the collection of all MI-LCQP-graphs.

**GNNs for MI-LCQP-graphs** are constructed using the same framework as those for LCQP-graphs, differing only in the input feature $w_j$ as defined above. We use $\mathcal{F}_{\text{MI-LCQP}}$ and $\mathcal{F}_{\text{MI-LCQP}}^W$ to denote the GNN classes for MI-LCQP-graphs with graph-level and node-level output, respectively.

**Target mappings** for MI-LCQPs also similar to those in Definition 2.3. In particular, $\Phi_{\text{feas}}$ and $\Phi_{\text{obj}}$ are defined as in Definition 2.3, while the optimal solution mapping $\Phi_{\text{sol}}$ can only be defined on a subset of the class of feasible and bounded MI-LCQPs, as discussed in Appendix C.

### 4.1. GNNs cannot universally represent MI-LCQPs

In this subsection, we answer the question (1) for MI-LCQP. When integer variables are introduced, the situation changes. Particularly, we present some counter-examples illustrating the fundamental limitation of GNNs in this context.

**Proposition 4.1.** *There exist two MI-LCQP problems, with one being feasible and the other being infeasible, such that their graphs are indistinguishable by any GNN in $\mathcal{F}_{\text{MI-CLQP}}$.*

**Proposition 4.2.** *There exist two feasible MI-LCQP problems, with different optimal objective values, such that their graphs are indistinguishable by any GNN in $\mathcal{F}_{\text{MI-CLQP}}$.*

**Proposition 4.3.** *There exist two feasible MI-LCQP problems with the same optimal objectives but disjoint optimal solution sets, such that their graphs are indistinguishable by any GNN in $\mathcal{F}_{\text{MI-CLQP}}^W$.*

It is indicated that for some MI-LCQP data distribution, it is *impossible* to train a GNN to predict MI-LCQP properties, *regardless of the GNN's size*. Particularly, one can

choose the uniform distribution over pairs of instances satisfying Propositions 4.1, 4.2, and 4.3: any GNN making good approximation on one instance must fail on the other.

The detailed proofs are provided in Appendix B. Here we present a pair of MI-LCQPs that prove Proposition 4.3.

**A counterexample.** We modify the two LCQPs in (5) and (6) into two MI-LCQPs by introducing integer constraints on all variables: specifically, $x_j \in \mathbb{Z}$ for $i \in \{1, \cdots, 7\}$. Consequently, the node attributes $w_j$ are updated to $w_j = (1, 0, 3, 1)$, where the last entry, $1$, indicates the integral constraint $\delta_I(j) = 1$. All other components of the graphs remain unchanged, as described in Section 3. With this modification, the node attributes $w_j$ remain identical for all $j \in W$. Since all other graph components are unchanged, *the same argument from Section 3 applies: any GNNs cannot distinguish the two MI-LCQPs.*

However, the mixed-integer setting differs from the continuous one: the two problems no longer share the same solution $(\frac{6}{7}, \frac{6}{7}, \frac{6}{7}, \frac{6}{7}, \frac{6}{7}, \frac{6}{7}, \frac{6}{7})$. Instead, they have *disjoint* optimal solution sets. The first instance has an *unique* feasible (and thus optimal) solution $(3, 3, 0, 0, 0, 0, 0)$, while in the second instance, it is $(2, 2, 2, 0, 0, 0, 0)$.

### 4.2. GNNs can represent particular types of MI-LCQPs

We have shown a fundamental limitation of GNNs to represent MI-LCQP in general. A natural question arise: *Whether we can identify a subset of $\mathcal{G}_{\text{MI-LCQP}}$ on which it is possible to train reliable GNNs.* To address this, we need to gain a better understanding for the separation power of GNNs, or equivalently, of the WL test, according to the discussion following Theorem 3.4. We state in Algorithm 1 the WL test for MI-LCQP-graphs associated to $\mathcal{F}_{\text{MI-LCQP}}$ or $\mathcal{F}_{\text{MI-LCQP}}^W$.

---

**Algorithm 1** The WL test (Ref. to App. D for an example)

---

**Require:** $G = (V, W, A, Q, H_V, H_W)$, and iteration limit $L$.
1: Initialization: $C_i^{0,V} = \text{hash}(v_i)$, and $C_j^{0,W} = \text{hash}(w_j)$.
2: **for** $l = 1, 2, \cdots, L$ **do**
3: $\quad C_i^{l,V} = \text{hash}\left(C_i^{l-1,V}, \{\!\{(C_j^{l-1,W}, A_{ij})\}\!\}_{j \in \mathcal{N}_i^W}\right)$.
4: $\quad C_j^{l,W} = \text{hash}\big(C_j^{l-1,W}, \{\!\{(C_i^{l-1,V}, A_{ij})\}\!\}_{i \in \mathcal{N}_j^V},$
$\quad\quad\quad\quad\quad\quad\quad \{\!\{(C_{j'}^{l-1,W}, Q_{jj'})\}\!\}_{j' \in \mathcal{N}_j^W}\big)$.
5: **end for**
6: **return** All vertex colors $\{\!\{C_i^{L,V}\}\!\}_{i=0}^m$, $\{\!\{C_j^{L,W}\}\!\}_{j=0}^n$.

---

Here, $C_i^{l,V}$ and $C_j^{l,W}$ are the colors of node $i \in V$ and node $j \in W$ at the $l$-th iteration. The "hash" function is any injective (collision-free) mapping, independent of $i, j$. In practice, standard built-in hash functions can be used.

The WL test mimics GNNs' iterative updates, replacing learnable mappings with hash functions. While hash functions cannot directly map a graph to desired outcomes, they effectively distinguish nodes and differentiate graphs.

Initially, each vertex is labeled a color by a hash function

according to its attributes ($v_i$ or $w_j$). At the $l$-th iteration, two vertices are of the same color if and only if at the $(l-1)$-th iteration, they have the same color and the same information aggregation from neighbors. This is a *color refinement* procedure. One can have a **_partition_** of the vertex set $V \cup W$ at each iteration based on vertices' colors: two vertices are classified in the same class if and only if they are of the same color. Such a partition is strictly refined in the first $\mathcal{O}(m+n)$ iterations and will *remain stable or unchanged* afterward, see e.g., Berkholz et al. (2017).

**Revisit the counterexample in Section 4.1:** As analyzed earlier, applying the WL test to the two MI-LCQPs results in all variable nodes $W$ having identical colors (red nodes), meaning they belong to the same class in the WL test's final stable partition. Nodes in the same class of this partition will always share identical attributes across all layers of any GNNs, and vice versa, because the color refinement process in Algorithm 1 mirrors GNNs' update mechanisms. However, while the desired GNN output is $(3, 3, 0, 0, 0, 0, 0)$, GNNs cannot differentiate between these variables, resulting in outputs where $y_1 = \cdots = y_7$. This contradiction is the core reason for GNNs' failure on these counterexamples.

To address this issue, we propose to: (I) Restrict the focus to MI-LCQPs where each node in the graph representation has a unique color (i.e., every class in the partition contains exactly one node), ensuring GNNs can separate all nodes. or (II) Allow partitions with classes containing multiple nodes but impose additional conditions within each class to ensure the desired output does not rely on distinguishing between these nodes. Therefore, we define as follows.

**Definition 4.4** (GNN-friendly MI-LCQPs). Consider a MI-LCQP problem $G_{\text{MI-LCQP}} \in \mathcal{G}_{\text{MI-LCQP}}^{m,n}$ and apply the WL test to obtain the final stable partition $(\mathcal{I}, \mathcal{J})$, where $\mathcal{I} = \{I_1, I_2, \ldots, I_s\}$ partitions $V = \{1, 2, \ldots, m\}$ and $\mathcal{J} = \{J_1, J_2, \ldots, J_t\}$ partitions $W = \{1, 2, \ldots, n\}$. Then,

- $G_{\text{MI-LCQP}}$ is called "GNN-solvable" if $t = n$ and $|J_1| = |J_2| = \cdots = |J_n| = 1$, i.e., all vertices in $W$ have different colors. We use $\mathcal{G}_{\text{solvable}}^{m,n} \subset \mathcal{G}_{\text{MI-LCQP}}^{m,n}$ to denote the collection of all GNN-solvable MI-LCQPs.

- $G_{\text{MI-LCQP}}$ is called "GNN-analyzable" if: (1) For any $p \in \{1, 2, \ldots, s\}$ and $q \in \{1, 2, \ldots, t\}$, $A_{ij}$ is constant across all $i \in I_p, j \in J_q$. (2) For any $q, q' \in \{1, 2, \ldots, t\}$, $Q_{jj'}$ is constant across $j \in J_q, j' \in J_{q'}$. The set of all such MI-LCQPs is denoted by $\mathcal{G}_{\text{analyzable}}^{m,n} \subset \mathcal{G}_{\text{MI-LCQP}}^{m,n}$.

GNN-solvability extends "unfoldability" in the context of MILP (Chen et al., 2023b), and GNN-analyzability extends "MP-tractability" (Chen et al., 2024). With such definitions, we can establish GNNs' expressive power for MI-LCQPs.

**Assumption 4.5.** $\mathbb{P}$ is a Borel regular probability measure defined on the space of MI-LCQP-graphs $\mathcal{G}_{\text{MI-LCQP}}^{m,n}$.

**Theorem 4.6.** *For any $\mathbb{P}$ satisfying Assumption 4.5 and $\mathbb{P}[G_{\text{MI-LCQP}} \in \mathcal{G}_{\text{analyzable}}^{m,n}] = 1$, and for any $\epsilon > 0$, there*

*exists $F \in \mathcal{F}_{\text{MI-LCQP}}$ such that*

$$\mathbb{P}\left[\mathbb{I}_{F(G_{\text{MI-LCQP}}) > \frac{1}{2}} \neq \Phi_{\text{feas}}(G_{\text{MI-LCQP}})\right] < \epsilon, \qquad (7)$$

*and there exists $F_1 \in \mathcal{F}_{\text{MI-LCQP}}$ such that*

$$\mathbb{P}\left[\mathbb{I}_{F_1(G_{\text{MI-LCQP}}) > \frac{1}{2}} \neq \mathbb{I}_{\Phi_{\text{obj}}(G_{\text{MI-LCQP}}) \in \mathbb{R}}\right] < \epsilon. \qquad (8)$$

*Additionally, if $\mathbb{P}[\Phi_{\text{obj}}(G_{\text{MI-LCQP}}) \in \mathbb{R}] = 1$, for any $\epsilon, \delta > 0$, there exists $F_2 \in \mathcal{F}_{\text{MI-LCQP}}$ such that*

$$\mathbb{P}\left[|F_2(G_{\text{MI-LCQP}}) - \Phi_{\text{obj}}(G_{\text{MI-LCQP}})| > \delta\right] < \epsilon. \quad (9)$$

To extend these results to predicting optimal solutions, we need to assume the MI-LCQPs have an optimal solution. We denote $\mathcal{G}_{\text{sol}}^{m,n}$ as the set of MI-LCQPs with an optimal solution. The assumption is expressed as $G_{\text{MI-LCQP}} \in \mathcal{G}_{\text{sol}}^{m,n}$.

**Theorem 4.7.** *For any $\mathbb{P}$ satisfying Assumption 4.5 and $\mathbb{P}[G_{\text{MI-LCQP}} \in \mathcal{G}_{\text{sol}}^{m,n} \cap \mathcal{G}_{\text{solvable}}^{m,n}] = 1$, and for any $\epsilon, \delta > 0$, there exists $F_W \in \mathcal{F}_{\text{MI-LCQP}}^W$ such that*

$$\mathbb{P}\left[\|F_W(G_{\text{MI-LCQP}}) - \Phi_{\text{sol}}(G_{\text{MI-LCQP}})\| > \delta\right] < \epsilon.$$

Theorems 4.6 and 4.7 indicate the subsets of MI-LCQPs where GNNs can succeed: For *GNN-analyzable* MI-LCQPs, GNNs can approximate their feasibility and optimal objective; For *GNN-solvable* ones, GNNs can approximate their optimal solutions. Proofs are available in Appendix C.

### 4.3. Discussions of "friendly" MI-LCQPs

To better illustrate the practical implications of Theorems 4.6 and 4.7, we make some discussions of GNN-analyzability and GNN-solvability here.

**Analyzability vs Solvability.** While all GNN-solvable MI-LCQPs must be GNN-analyzable, not all GNN-analyzable problems are necessarily GNN-solvable. Related discussions, proofs and examples are detailed in Appendix D.

Since solvability is a stronger condition than analyzability, when a problem is GNN-solvable, GNNs can approximate not only the optimal solution (Theorem 4.7) but also the feasibility and the optimal objective (Theorem 4.6). In this sense, learning the solution mapping is a stricter requirement than learning the feasibility and optimal objective.

**Frequency of friendly instances.** In practice, the frequency of GNN-analyzable and GNN-solvable instances largely depends on the **_the level of symmetry_** in the dataset. Here, two variables labeled with the same color by the WL test are said to be symmetric. When there is symmetry in a MI-LCQP, it becomes GNN-insolvable; and higher symmetry increases the risk of being GNN-inanalyzable. Examples provided by (5) and (6) admit strong symmetry across all variables, making them neither GNN-analyzable nor GNN-solvable. Fortunately, *GNN-solvable and GNN-analyzable instances make up the majority of the MI-LCQP set*, under specific distributional assumptions (see e.g. Proposition D.3). This explains the observed success of GNNs for QPs in the existing literature and addresses the gap between theoretical limitations

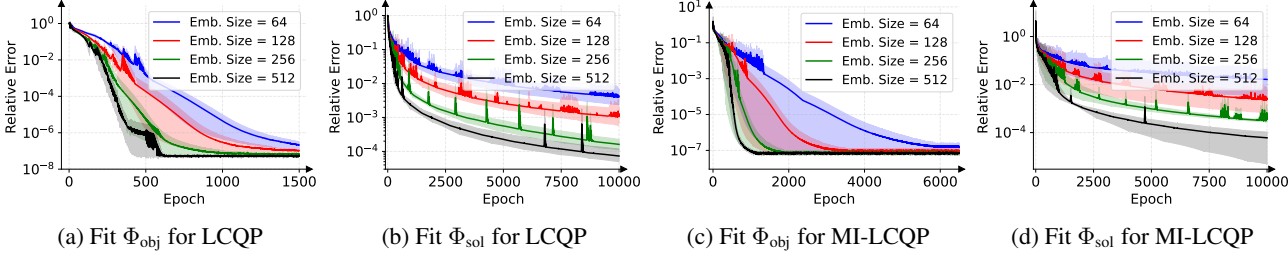

(a) Fit $\Phi_{\text{obj}}$ for LCQP     (b) Fit $\Phi_{\text{sol}}$ for LCQP     (c) Fit $\Phi_{\text{obj}}$ for MI-LCQP     (d) Fit $\Phi_{\text{sol}}$ for MI-LCQP

*Figure 2.* Relative errors when training GNNs to fit $\Phi_{\text{obj}}$ and $\Phi_{\text{sol}}$ for LCQP (2a-2b) and MI-LCQP (2c-2d) with various embedding sizes.

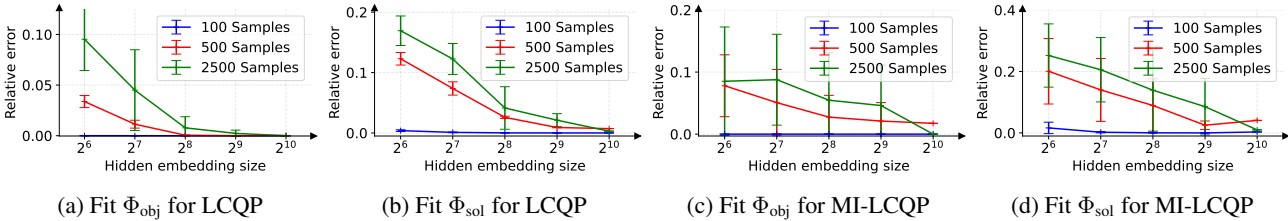

(a) Fit $\Phi_{\text{obj}}$ for LCQP     (b) Fit $\Phi_{\text{sol}}$ for LCQP     (c) Fit $\Phi_{\text{obj}}$ for MI-LCQP     (d) Fit $\Phi_{\text{sol}}$ for MI-LCQP

*Figure 3.* Validation on a larger scale. The figures illustrate the relative errors for various combinations of embedding sizes and numbers of training samples. We can achieve near zero errors when the GNN is large enough.

and empirical success discussed in Section 1. The dataset used in Section 5 consists entirely of GNN-solvable and GNN-analyzable instances. However, in some challenging, artificially collected or created datasets, such as QPlib (Furini et al., 2019), it's possible to be GNN-insolvable or even GNN-inanalyzable (see in Appendix D.2 for details).

**Numerical verification.** While GNN-analyzability and GNN-solvability depend on the dataset, they can be efficiently verified in practice. One may first apply Algorithm 1, which requires at most $\mathcal{O}(m + n)$ iterations. The single-iteration complexity is bounded by the number of edges in the graph (Shervashidze et al., 2011), which, in our context, is the number of nonzeros in matrices $A$ and $Q$: $\text{nnz}(A) + \text{nnz}(Q)$. Therefore, Algorithm 1's complexity is $\mathcal{O}((m + n) \cdot (\text{nnz}(A) + \text{nnz}(Q)))$. Afterward, both conditions can be directly verified using Definition 4.4.

Therefore, these criteria provide practitioners with a practical tool to evaluate datasets before applying GNNs. Additionally, if challenges arise during GNN training on MI-LCQPs, verifying them can help identify potential issues.

## 5. Numerical experiments

We present numerical experiments in this section.[3]

**Numerical validation of GNNs' expressive power.** We train GNNs to fit $\Phi_{\text{obj}}$ or $\Phi_{\text{sol}}$ for LCQP or MI-LCQP instances.[4] For both LCQP and MI-LCQP, we randomly generate 100 instances, each of which contains 10 constraints

and 50 variables. The generated MI-LCQPs are all GNN-solvable and GNN-analyzable with probability one. Details on the data generation and training schemes can be found in Appendix G. We train four GNNs with four different embedding sizes and record their relative errors[5] averaged on all instances during training. The results are reported in Figure 2. We can see that GNNs can fit $\Phi_{\text{obj}}$ and $\Phi_{\text{sol}}$ well for both LCQP and MI-LCQP. These results validate Theorems 3.3,3.4,4.6 and 4.7. We also observe that a larger embedding size increases the capacity of a GNN, resulting in not only lower final errors but also faster convergence.

**Validation on a larger scale.** To further validate the theorems, we expand the number of problem instances to 500 and 2,500. The results, reported in Figure 3, show that GNN achieves near-zero fitting errors when it has a large enough embedding size and thus enough capacity for approximation, which directly validate Theorems 3.3,3.4,4.6 and 4.7.

**Various types of LCQP.** Besides the generic LCQP formulation (2), we also extend the numerical experiments to other types of optimization problems, namely portfolio optimization and support vector machine (SVM) following Jung et al. (2022). The results are reported in Figure 4. We can observe similar fitting behaviors as those in the generic LCQP experiments where the expressive power of GNNs increase as they become larger, evidenced by the fitting errors decreasing to near zero when the embedding size increases.

**Real dataset.** To further examine the universal approximation results on more realistic QP problems. For LCQP, we train GNNs on the Maros and Meszaros Problem Set (Maros

---

[3]Codes are available at `https://github.com/liujl11git/GNN-QP`.

[4]Since LCQP and MI-LCQP are linearly constrained, predicting feasibility falls to the case of LP and MILP in Chen et al. (2023a;b). Hence we omit the feasibility experiments.

[5]The relative error of a GNN $F_W$ on a single problem instance $G$ is defined as $\|F_W(G) - \Phi(G)\|_2 / \max(\|\Phi(G)\|_2, 1)$, where $\Phi$ could be either $\Phi_{\text{obj}}$ or $\Phi_{\text{sol}}$.

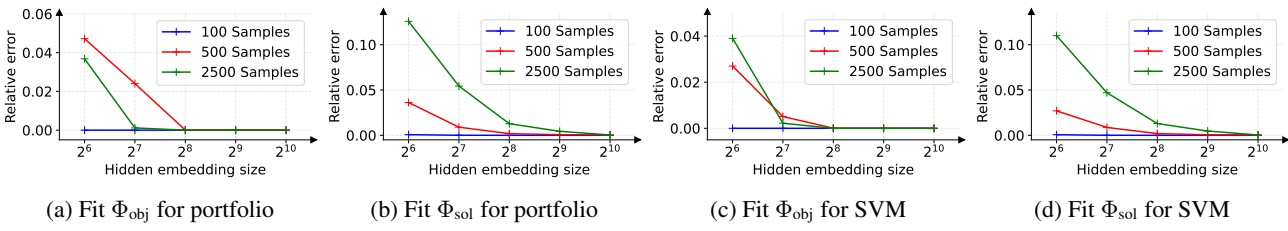

(a) Fit $\Phi_{obj}$ for portfolio    (b) Fit $\Phi_{sol}$ for portfolio    (c) Fit $\Phi_{obj}$ for SVM    (d) Fit $\Phi_{sol}$ for SVM

*Figure 4.* Various types of LCQP: portfolio optimization and SVM optimization problems. The figures illustrate the best relative errors achieved during training for various combinations of embedding sizes and numbers of training samples.

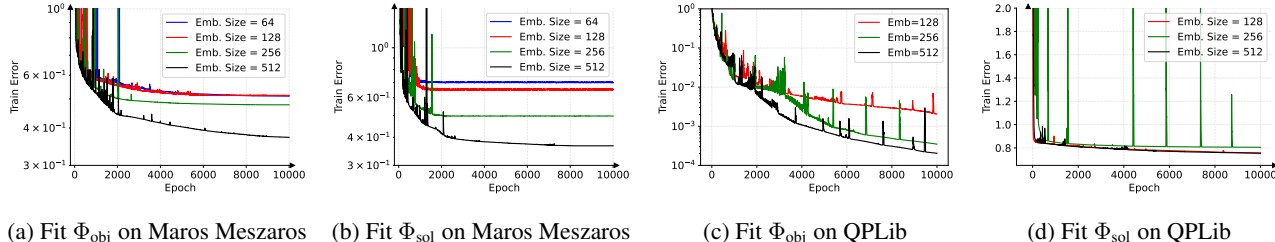

(a) Fit $\Phi_{obj}$ on Maros Meszaros    (b) Fit $\Phi_{sol}$ on Maros Meszaros    (c) Fit $\Phi_{obj}$ on QPLib    (d) Fit $\Phi_{sol}$ on QPLib

*Figure 5.* Training errors on real datasets: Convex LCQP (5a-5b) and MI-LCQP (5c-5d).

*Table 1.* Average solving times (milliseconds) of GNN and OSQP.

| Batch Size | - | 1 | 10 | 100 | 1,000 |
|---|---|---|---|---|---|
| OSQP | 4.48±3.62 | - | - | - | - |
| GNN | - | 53.62±16.72 | 5.37±1.87 | 0.504±0.142 | 0.089±0.002 |

& Mészáros, 1999), which contains 138 challenging convex LCQPs. The results are shown in Figure 5. We observe that while the broad range of numbers of instances in the Maros Meszaros test set caused numerical difficulties for training, GNNs can still be trained to fit the objectives and solutions to some extent. And we can observe similar tendency as in the synthesized experiments that the expressive power increases as the model capacity enlarges when we increase the embedding size. Details are deferred to Appendix G.

For MI-LCQP, we perform GNN training on 73 GNN-solvable instances from QPLib (Furini et al., 2019) that provide the optimal solutions and objectives. We train GNNs of embedding sizes 128, 256, and 512 to fit the objectives and solutions. The training errors we achieved are shown in Figures 5c-5d. GNNs can fit the objective values well and demonstrate the ability to fit solutions. The results show the model capacity improves as the model size increases.

**Computation complexity.** GNNs are superior over QP solvers in terms of running time, especially when we fully exploit parallel computing with GPU acceleration. To show this, we measure the average running time using OSQP (Stellato et al., 2020) and a trained GNN with different batch sizes over the 1,000 synthetic LCQP problems generated in the experiment above. We applied OSQP to solve all instances to a relative error of $10^{-3}$, which is slightly less accurate than the trained GNN (with an average relative error of $6.31 \times 10^{-4}$). The average solving times and standard

deviations are shown in Table 1. The sufficiently accelerated computation validates GNNs' capacity as a real-time QP solver or fast warm-start, numerically supporting the rationality of our theoretical study of GNNs for QPs.

**Generalization.** Besides investigating GNNs' expressive capacity, we also explore their generalization ability and observed positive results. As generalization is beyond this work's scope, the results are provided in Appendix G.

## 6. Conclusions and future directions

This paper establishes theoretical foundations for using GNNs to represent the feasibility, optimal objective value, and optimal solution of LCQPs and MI-LCQPs. We prove GNNs can universally approximate these properties for LCQPs and show this is in general not true for MI-LCQPs, except for specific subclasses we identify. Future research directions include studying the training dynamics, generalization behavior, and size requirements of GNNs. In particular, the algorithm-unrolling framework by Yang et al. (2024) provides a way to estimate GNN sizes, as it explicitly defines the number of parameters per layer, and prior work (e.g., (Li et al., 2024)) shows algorithm-unrolling is a structured GNN. These connections suggest that GNN complexity bounds for (MI-)LCQPs may be derived from the algorithm-unrolling literature. Another compelling direction for future work is to unify LP, LCQP, QCQP, and extensions on higher-order polynomial optimization via hypergraph GNNs.

## Impact Statement

This paper explores the expressive power of GNNs for QP tasks. Given the wide applications of QP in industry, our work could have broad societal impacts. However, as this study is primarily theoretical, we do not identify any specific societal impacts that must be specifically highlighted here.

## Acknowledgements

The work of Ziang Chen is supported in part by the National Science Foundation via grant DMS-2509011.

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

# A. Proofs for Section 3

In this appendix, we present the proofs for theorems in Section 3. The proofs will based on Weisfeiler-Lehman (WL) test and its separation power to distinguish LCQP problems with different properties (different feasibility, different optimal objective, or different optimal solution with the smallest $\ell_2$ norm).

The Weisfeiler-Lehman (WL) test (Weisfeiler & Leman, 1968) is a classical algorithm for the graph isomorphism problem. In particular, it implements color refinement on vertices by applying a hash function on the previous vertex color and aggregation of colors from neighbors, and identifies two graphs as isomorphic if their final color multisets are the same. It is worth noting that WL test may incorrectly identify two non-isomorphic graphs as isomorphic. We slightly modify the standard WL test, see Algorithm 2.

---

**Algorithm 2** The linear-form WL test

---

**Require:** A LCQP-graph $G = (V, W, A, Q, H_V, H_W)$ and iteration limit $L > 0$.

1: Initialize with $C_i^{0,V} = \text{hash}(v_i)$ and $C_j^{0,W} = \text{hash}(w_j)$.

2: **for** $l = 1, 2, \cdots, L$ **do**

3:    Refine the colors

$$C_i^{l,V} = \text{hash}\left(C_i^{l-1,V}, \sum_{j=1}^{n} A_{ij}\text{hash}\left(C_j^{l-1,W}\right)\right),$$

$$C_j^{l,W} = \text{hash}\left(C_j^{l-1,W}, \sum_{i=1}^{m} A_{ij}\text{hash}\left(C_i^{l-1,V}\right), \sum_{j'=1}^{n} Q_{jj'}\text{hash}\left(C_{j'}^{l-1,W}\right)\right).$$

4: **end for**

5: **return** The multisets containing all colors $\{\!\{C_i^{L,V}\}\!\}_{i=0}^{m}, \{\!\{C_j^{L,W}\}\!\}_{j=0}^{n}$.

---

Please note that, Algorithm 2 is a special case of Algorithm 1 and, as such, has weaker separation power. However, it is sufficient for distinguishing LCQPs with different key properties. Notably, proving universal approximation under a weaker WL test leads to an even *stronger* conclusion: if a weaker WL test (or equivalently, a less powerful GNN) can separate LCQPs, it automatically implies that these LCQPs can also be separated under a stronger WL test.

We define two equivalence relations as follows. Intuitively, LCQP-graphs in the same equivalence class will be identified as isomorphic by WL test, though they may be actually non-isomorphic.

**Definition A.1.** For two LCQP-graphs $G_{\text{LCQP}}, \hat{G}_{\text{LCQP}} \in \mathcal{G}_{\text{LCQP}}^{m,n}$, let $\{\!\{C_i^{L,V}\}\!\}_{i=0}^{m}, \{\!\{C_j^{L,W}\}\!\}_{j=0}^{n}$ and $\{\!\{\hat{C}_i^{L,V}\}\!\}_{i=0}^{m}, \{\!\{\hat{C}_j^{L,W}\}\!\}_{j=0}^{n}$ be color multisets output by Algorithm 2 on $G_{\text{LCQP}}$ and $\hat{G}_{\text{LCQP}}$.

1. We say $G_{\text{LCQP}} \sim \hat{G}_{\text{LCQP}}$ if $\{\!\{C_i^{L,V}\}\!\}_{i=0}^{m} = \{\!\{\hat{C}_i^{L,V}\}\!\}_{i=0}^{m}$ and $\{\!\{C_j^{L,W}\}\!\}_{j=0}^{n} = \{\!\{\hat{C}_j^{L,W}\}\!\}_{j=0}^{n}$ hold for all $L \in \mathbb{N}$ and all hash functions.

2. We say $G_{\text{LCQP}} \overset{W}{\sim} \hat{G}_{\text{LCQP}}$ if $\{\!\{C_i^{L,V}\}\!\}_{i=0}^{m} = \{\!\{\hat{C}_i^{L,V}\}\!\}_{i=0}^{m}$ and $C_j^{L,W} = \hat{C}_j^{L,W}, \forall j \in \{1, 2, \ldots, n\}$, for all $L \in \mathbb{N}$ and all hash functions.

Our main finding leading to the results in Section 3 is that, for LCQP-graphs in the same equivalence class, even if they are non-isomorphic, their optimal objective values and optimal solutions must be the same (up to a permutation perhaps).

**Theorem A.2.** *For any* $G_{\text{LCQP}}, \hat{G}_{\text{LCQP}} \in \mathcal{G}_{\text{LCQP}}^{m,n}$ *with* $Q, \hat{Q} \succeq 0$, *if* $G_{\text{LCQP}} \sim \hat{G}_{\text{LCQP}}$, *then* $\Phi_{\text{obj}}(G_{\text{LCQP}}) = \Phi_{\text{obj}}(\hat{G}_{\text{LCQP}})$.

**Theorem A.3.** *For any* $G_{\text{LCQP}}, \hat{G}_{\text{LCQP}} \in \mathcal{G}_{\text{LCQP}}^{m,n}$ *with* $Q, \hat{Q} \succeq 0$ *that are feasible and bounded, if* $G_{\text{LCQP}} \sim \hat{G}_{\text{LCQP}}$, *then there exists some permutation* $\sigma_W \in S_n$ *such that* $\Phi_{\text{sol}}(G_{\text{LCQP}}) = \sigma_W(\Phi_{\text{sol}}(\hat{G}_{\text{LCQP}}))$. *Furthermore, if* $G_{\text{LCQP}} \overset{W}{\sim} \hat{G}_{\text{LCQP}}$, *then* $\Phi_{\text{sol}}(G_{\text{LCQP}}) = \Phi_{\text{sol}}(\hat{G}_{\text{LCQP}})$.

We need the following lemma to prove Theorem A.2 and Theorem A.3.

**Lemma A.4.** *Suppose that $M \in \mathbb{R}^{n \times n}$ is a symmetric and positive semidefinite matrix and that $\mathcal{J} = \{J_1, J_2, \ldots, J_t\}$ is a partition of $\{1, 2, \ldots, n\}$ satisfying that for any $q, q' \in \{1, 2, \ldots, t\}$, $\sum_{j' \in J_{q'}} M_{jj'}$ is a constant over $j \in J_q$. For any $x \in \mathbb{R}^n$, it holds that*

$$\frac{1}{2} x^\top M x \geq \frac{1}{2} \hat{x}^\top M \hat{x}, \tag{10}$$

*where $\hat{x} \in \mathbb{R}^n$ is defined via $\hat{x}_j = y_q = \frac{1}{|J_q|} \sum_{j' \in J_q} x_{j'}$ for $j \in J_q$.*

*Proof.* Fix $x \in \mathbb{R}^n$ and consider the problem

$$\min_{z \in \mathbb{R}^n} \frac{1}{2} z^\top M z, \quad \text{s.t.} \quad \sum_{j \in J_q} z_j = \sum_{j \in J_q} x_j, \ q = 1, 2, \ldots, t, \tag{11}$$

which is a convex program. The Lagrangian is given by

$$\mathcal{L}(z, \lambda) = \frac{1}{2} z^\top M z - \sum_{q=1}^{t} \lambda_q \left( \sum_{j \in J_q} z_j - \sum_{j \in J_q} x_j \right).$$

It can be computed that

$$\frac{\partial}{\partial z_j} \mathcal{L}(z, \lambda) = \sum_{j'=1}^{n} M_{jj'} z_{j'} - \lambda_q, \quad j \in J_q,$$

and

$$\frac{\partial}{\partial \lambda_q} \mathcal{L}(z, \lambda) = \sum_{j \in J_q} x_j - \sum_{j \in J_q} z_j,$$

It is clear that

$$\frac{\partial}{\partial \lambda_q} \mathcal{L}(\hat{x}, \lambda) = \sum_{j \in J_q} x_j - \sum_{j \in J_q} \hat{x}_j = 0,$$

by the definition of $\hat{x}$. Furthermore, consider any fixed $q \in \{1, 2, \ldots, t\}$ and we have for any $j \in J_q$ that

$$\frac{\partial}{\partial z_j} \mathcal{L}(\hat{x}, \lambda) = \sum_{q'=1}^{t} y_{q'} \sum_{j' \in J_{q'}} M_{jj'} - \lambda_q = 0,$$

if $\lambda_q = \sum_{q'=1}^{t} y_{q'} \sum_{j' \in J_{q'}} M_{jj'}$ that is independent in $j \in q$ since $\sum_{j' \in J_{q'}} M_{jj'}$ is constant over $j \in J_q$ for any $q' \in \{1, 2, \ldots, t\}$. Since the problem (11) is convex and the first-order optimality condition is satisfied at $\hat{x}$, we can conclude that $\hat{x}$ is a minimizer of (11), which implies (10). $\square$

*Proof of Theorem A.2.* Let $G_{\text{LCQP}}$ and $\hat{G}_{\text{LCQP}}$ be the LCQP-graphs associated to (2) and

$$\min_{x \in \mathbb{R}^n} \frac{1}{2} x^\top \hat{Q} x + \hat{c}^\top x, \quad \text{s.t.} \ \hat{A} x \ \hat{\circ} \ \hat{b}, \ \hat{l} \leq x \leq \hat{u}, \tag{12}$$

Suppose that there are no collisions of hash functions or their linear combinations when applying the WL test to $G_{\text{LCQP}}$ and $\hat{G}_{\text{LCQP}}$ and there are no strict color refinements in the $L$-th iteration. Since $G_{\text{LCQP}} \sim \hat{G}_{\text{LCQP}}$, after performing some permutation, there exist $\mathcal{I} = \{I_1, I_2, \ldots, I_s\}$ and $\mathcal{J} = \{J_1, J_2, \ldots, J_t\}$ that are partitions of $\{1, 2, \ldots, m\}$ and $\{1, 2, \ldots, n\}$, respectively, such that the followings hold:

- $C_i^{L,V} = C_{i'}^{L,V}$ if and only if $i, i' \in I_p$ for some $p \in \{1, 2, \ldots, s\}$.

- $C_i^{L,V} = \hat{C}_{i'}^{L,V}$ if and only if $i, i' \in I_p$ for some $p \in \{1, 2, \ldots, s\}$.

- $\hat{C}_i^{L,V} = \hat{C}_{i'}^{L,V}$ if and only if $i, i' \in I_p$ for some $p \in \{1, 2, \ldots, s\}$.

- $C_j^{L,W} = C_{j'}^{L,W}$ if and only if $j, j' \in J_q$ for some $q \in \{1, 2, \ldots, t\}$.

- $C_j^{L,W} = \hat{C}_{j'}^{L,W}$ if and only if $j, j' \in J_q$ for some $q \in \{1, 2, \ldots, t\}$.

- $\hat{C}_j^{L,W} = \hat{C}_{j'}^{L,W}$ if and only if $j, j' \in J_q$ for some $q \in \{1, 2, \ldots, t\}$.

Since there are no collisions, we have from the vertex color initialization that

- $v_i = (b_i, \circ_i) = \hat{v}_i = (\hat{b}_i, \hat{\circ}_i)$ and is constant over $i \in I_p$ for any $p \in \{1, 2, \ldots, s\}$.

- $w_j = (c_j, l_j, u_j) = \hat{w}_j = (\hat{c}_j, \hat{l}_j, \hat{u}_j)$ and is constant over $j \in J_q$ for any $q \in \{1, 2, \ldots, t\}$.

For any $p \in \{1, 2, \ldots, s\}$ and any $i, i' \in I_p$, one has

$$
\begin{aligned}
C_i^{L,V} = C_{i'}^{L,V} &\implies \sum_{j \in W} A_{ij} \mathrm{hash}\left(C_j^{L-1,W}\right) = \sum_{j \in W} A_{i'j} \mathrm{hash}\left(C_j^{L-1,W}\right) \\
&\implies \sum_{j \in W} A_{ij} \mathrm{hash}\left(C_j^{L,W}\right) = \sum_{j \in W} A_{i'j} \mathrm{hash}\left(C_j^{L,W}\right) \\
&\implies \sum_{j \in J_q} A_{ij} = \sum_{j \in J_q} A_{i'j}, \quad \forall q \in \{1, 2, \ldots, t\}.
\end{aligned}
$$

One can obtain similar conclusions from $C_i^{L,V} = \hat{C}_{i'}^{L,V}$ and $\hat{C}_i^{L,V} = \hat{C}_{i'}^{L,V}$, and hence conclude that

- For any $p \in \{1, 2, \ldots, s\}$ and $q \in \{1, 2, \ldots, t\}$, $\sum_{j \in J_q} A_{ij} = \sum_{j \in J_q} \hat{A}_{ij}$ and is constant over $i \in I_p$.

Similarly, the followings also hold:

- For any $p \in \{1, 2, \ldots, s\}$ and $q \in \{1, 2, \ldots, t\}$, $\sum_{i \in I_p} A_{ij} = \sum_{i \in I_p} \hat{A}_{ij}$ and is constant over $j \in J_q$.

- For any $q, q' \in \{1, 2, \ldots, t\}$, $\sum_{j' \in J_{q'}} Q_{jj'} = \sum_{j' \in J_{q'}} \hat{Q}_{jj'}$ and is constant over $j \in J_q$.

If $G_{\mathrm{LCQP}}$ or (2) is infeasible, then $\Phi_{\mathrm{obj}}(G_{\mathrm{LCQP}}) = +\infty$ and clearly $\Phi_{\mathrm{obj}}(G_{\mathrm{LCQP}}) \geq \Phi_{\mathrm{obj}}(\hat{G}_{\mathrm{LCQP}})$. If (2) is feasible, let $x \in \mathbb{R}^n$ be any feasible solution to (2) and define $\hat{x} \in \mathbb{R}^n$ via $\hat{x}_j = y_q = \frac{1}{|J_q|} \sum_{j' \in J_q} x_{j'}$ for $j \in J_q$. By the proofs of Lemma B.2 and Lemma B.3 in Chen et al. (2023a), we know that $\hat{x}$ is a feasible solution to (12) and $c^\top x = \hat{c}^\top \hat{x}$. In addition, we have

$$
\begin{aligned}
\frac{1}{2} x^\top Q x \overset{(10)}{\geq} \frac{1}{2} \hat{x}^\top Q \hat{x} &= \frac{1}{2} \sum_{q,q'=1}^{t} \sum_{j \in J_q} \sum_{j' \in J_{q'}} \hat{x}_j Q_{jj'} \hat{x}_{j'} = \frac{1}{2} \sum_{q,q'=1}^{t} y_q y_{q'} \sum_{j' \in J_{q'}} Q_{jj'} \\
&= \frac{1}{2} \sum_{q,q'=1}^{t} y_q y_{q'} \sum_{j' \in J_{q'}} \hat{Q}_{jj'} = \frac{1}{2} \sum_{q,q'=1}^{t} \sum_{j \in J_q} \sum_{j' \in J_{q'}} \hat{x}_j \hat{Q}_{jj'} \hat{x}_{j'} = \frac{1}{2} \hat{x}^\top \hat{Q} \hat{x},
\end{aligned}
$$

which then implies that

$$
\frac{1}{2} x^\top Q x + c^\top x \geq \frac{1}{2} \hat{x}^\top \hat{Q} \hat{x} + \hat{c}^\top \hat{x},
$$

and hence that $\Phi_{\mathrm{obj}}(G_{\mathrm{LCQP}}) \geq \Phi_{\mathrm{obj}}(\hat{G}_{\mathrm{LCQP}})$. Till now we have proved $\Phi_{\mathrm{obj}}(G_{\mathrm{LCQP}}) \geq \Phi_{\mathrm{obj}}(\hat{G}_{\mathrm{LCQP}})$ regardless of the feasibility of $G_{\mathrm{LCQP}}$. The reverse direction $\Phi_{\mathrm{obj}}(G_{\mathrm{LCQP}}) \leq \Phi_{\mathrm{obj}}(\hat{G}_{\mathrm{LCQP}})$ is also true and we can conclude that $\Phi_{\mathrm{obj}}(G_{\mathrm{LCQP}}) = \Phi_{\mathrm{obj}}(\hat{G}_{\mathrm{LCQP}})$. $\square$

*Proof of Theorem A.3.* Under the same setting as in the proof of Theorem A.2, the results can be proved. We present the proof here.

Let $x \in \mathbb{R}^n$ be the optimal solution to (2) with the smallest $\ell_2$-norm, and let $\hat{x} \in \mathbb{R}^n$ be defined as in the proof of Theorem A.2. By the arguments in the proof of Theorem A.2, $\hat{x}$ is an optimal solution to (12). In particular, $\hat{x}$ is also an optimal solution to (2) since one can set $(\hat{A}, \hat{b}, \hat{c}, \hat{Q}, \hat{l}, \hat{u}, \hat{\circ}) = (A, b, c, Q, l, u, \circ)$. Therefore, by the minimality of $\|x\|^2$, we have that

$$\|x\|^2 \leq \|\hat{x}\|^2 = \sum_{q=1}^{t} \sum_{j \in J_q} \hat{x}_j^2 = \sum_{q=1}^{t} |J_q| \left( \frac{1}{|J_q|} \sum_{j \in J_q} x_j \right)^2 \leq \sum_{q=1}^{t} \sum_{j \in J_q} x_j^2 = \|x\|^2,$$

which implies that $x_j$ is a constant in $j \in J_q$ and $x = \hat{x}$. Thus, $x$ is also an optimal solution to (12).

Let $x' \in \mathbb{R}^n$ be the optimal solution to (12) with the smallest $\ell_2$-norm. Then $\|x'\| \leq \|\hat{x}\| = \|x\|$ and the reverse direction $\|x\| \leq \|x'\|$ is also true, which implies that $\|x\| = \|x'\|$. Therefore, we have $x = x'$ by the uniqueness of the optimal solution with the smallest $\ell_2$-norm.

Noticing that the above arguments are made after permuting vertices in $V$ and $W$, we can conclude that $\Phi_{\text{sol}}(G_{\text{LCQP}}) = \sigma_W(\Phi_{\text{sol}}(\hat{G}_{\text{LCQP}}))$ for some $\sigma_W \in S_n$. Additionally, if $G_{\text{LCQP}} \overset{W}{\sim} \hat{G}_{\text{LCQP}}$, then there is no need to perform the permutation on $W$ and we have $\Phi_{\text{sol}}(G_{\text{LCQP}}) = \Phi_{\text{sol}}(\hat{G}_{\text{LCQP}})$. $\qquad\square$

**Corollary A.5.** *For any $G_{\text{LCQP}} \in \mathcal{G}_{\text{LCQP}}^{m,n}$ that is feasible and bounded and any $j, j' \in \{1, 2, \ldots, n\}$, if $C_j^{L,W} = C_{j'}^{L,W}$ holds for all $L \in \mathbb{N}_+$ and all hash functions, then $\Phi_{\text{sol}}(G_{\text{LCQP}})_j = \Phi_{\text{sol}}(G_{\text{LCQP}})_{j'}$.*

*Proof.* Let $\hat{G}_{\text{LCQP}}$ be the LCQP-graph obtained from $G_{\text{LCQP}}$ by relabeling $j$ as $j'$ and relabeling $j'$ as $j$. By Theorem A.3, we have $\Phi_{\text{sol}}(G_{\text{LCQP}}) = \Phi_{\text{sol}}(\hat{G}_{\text{LCQP}})$, which implies $\Phi_{\text{sol}}(G_{\text{LCQP}})_j = \Phi_{\text{sol}}(\hat{G}_{\text{LCQP}})_j = \Phi_{\text{sol}}(G_{\text{LCQP}})_{j'}$. $\qquad\square$

It is well-known from previous literature that the separation power of GNNs is equivalent to that of WL test and that GNNs can universally approximate any continuous function whose separation is not stronger than that of WL test; see e.g. Chen et al. (2023a); Xu et al. (2019); Azizian & Lelarge (2021); Geerts & Reutter (2022). We have established in Theorem A.2, Theorem A.3, and Corollary A.5 that the separation power of $\Phi_{\text{obj}}$ and $\Phi_{\text{sol}}$ is upper bounded by the WL test (Algorithm 2) that shares the same information aggregation mechanism as the GNNs in $\mathcal{F}_{\text{LCQP}}$ and $\mathcal{F}_{\text{LCQP}}^W$. Theorem 3.3 and Theorem 3.4 will be proved following this idea.

*Proof of Theorem 3.3.* Theorem 3.3 can be proved based on Theorem A.2.

The separation power of GNNs is equivalent to that of the WL test, i.e., for any $G_{\text{LCQP}}, \hat{G}_{\text{LCQP}} \in \mathcal{G}_{\text{LCQP}}^{m,n}$ with $Q, \hat{Q} \succeq 0$,

$$G_{\text{LCQP}} \sim \hat{G}_{\text{LCQP}} \iff F(G_{\text{LCQP}}) = F(\hat{G}_{\text{LCQP}}), \ \forall \, F \in \mathcal{F}_{\text{LCQP}}, \tag{13}$$

which combined with Theorem A.2 leads to that

$$F(G_{\text{LCQP}}) = F(\hat{G}_{\text{LCQP}}), \ \forall \, F \in \mathcal{F}_{\text{LCQP}} \implies \Phi_{\text{obj}}(G_{\text{LCQP}}) = \Phi_{\text{obj}}(\hat{G}_{\text{LCQP}}), \tag{14}$$

indicating that the separation power of $\mathcal{F}_{\text{LCQP}}$ is upper bounded by that of $\Phi_{\text{obj}}$.

The indicator function $\mathbb{I}_{\Phi_{\text{obj}}(\cdot) \in \mathbb{R}} : \mathcal{G}_{\text{LCQP}}^{m,n} \to \{0, 1\} \subset \mathbb{R}$ is measurable, i.e., $\Phi_{\text{obj}}^{-1}(0)$ and $\Phi_{\text{obj}}^{-1}(0)$ are both Lebesgue measurable, and hence by Lusin's theorem, there exists a compact and permutation-invariant subspace $X \subset \mathcal{G}_{\text{LCQP}}^{m,n}$ such that $\mathbb{P}[\mathcal{G}_{\text{LCQP}}^{m,n} \backslash X] < \epsilon$ and that $\mathbb{I}_{\Phi_{\text{obj}}(\cdot) \in \mathbb{R}}$ restricted on $X$ is continuous. Therefore, by the Stone-Weierstrass theorem and (14), we have that there exists $F_1 \in \mathcal{F}_{\text{LCQP}}$ satisfying

$$\sup_{G_{\text{LCQP}} \in X} \left| F_1(G_{\text{LCQP}}) - \mathbb{I}_{\Phi_{\text{obj}}(G_{\text{LCQP}}) \in \mathbb{R}} \right| < \frac{1}{2}$$

Therefore, it holds that

$$\mathbb{P}\left[ \mathbb{I}_{F_1(G_{\text{LCQP}}) > \frac{1}{2}} \neq \mathbb{I}_{\Phi_{\text{obj}}(G_{\text{LCQP}}) \in \mathbb{R}} \right] \leq \mathbb{P}\left[ \mathcal{G}_{\text{LCQP}}^{m,n} \backslash X \right] < \epsilon,$$

which proves (3). Additionally, (4) can be proved by applying similar arguments to $\Phi_{\text{obj}} : \Phi_{\text{obj}}^{-1}(\mathbb{R}) \to \mathbb{R}$, where $\Phi_{\text{obj}}^{-1}(\mathbb{R}) \subset \mathcal{G}_{\text{LCQP}}^{m,n}$ is the collection of feasible and bounded $G_{\text{LCQP}} \in \mathcal{G}_{\text{LCQP}}^{m,n}$. $\qquad\square$

*Proof of Theorem 3.4.* Theorem 3.4 can be proved based on Theorem A.3 and Corollary A.5.

In addition to (13), it can be proved that the separation powers of GNNs and the WL test are equivalent in the following sense:

- For any $G_{\text{LCQP}}, \hat{G}_{\text{LCQP}} \in \mathcal{G}_{\text{LCQP}}^{m,n}$, $G_{\text{LCQP}} \overset{W}{\sim} \hat{G}_{\text{LCQP}}$ if and only if $F_W(G_{\text{LCQP}}) = F_W(\hat{G}_{\text{LCQP}})$ for all $F_W \in \mathcal{F}_{\text{LCQP}}^W$.

- For any $G_{\text{LCQP}} \in \mathcal{G}_{\text{LCQP}}^{m,n}$ and any $j, j' \in W$, $C_j^{L,W} = C_{j'}^{L,W}$ for any $L \in \mathbb{N}$ and any hash function if and only if $F_W(G_{\text{LCQP}})_j = F_W(G_{\text{LCQP}})_{j'}$ for all $F_W \in \mathcal{F}_{\text{LCQP}}^W$.

Therefore, with Theorem A.3 and Corollary A.5, the separation power of GNNs is upper bounded by that of $\Phi_{\text{sol}}$ in the following sense that for any $G_{\text{LCQP}}, \hat{G}_{\text{LCQP}} \in \mathcal{G}_{\text{LCQP}}^{m,n}$ with $Q, \hat{Q} \succeq 0$ and any $j, j' \in W$,

- $F(G_{\text{LCQP}}) = F(\hat{G}_{\text{LCQP}})$, $\forall\, F \in \mathcal{F}_{\text{LCQP}}$ implies $\Phi_{\text{sol}}(G_{\text{LCQP}}) = \sigma_W(\Phi_{\text{sol}}(\hat{G}_{\text{LCQP}}))$ for some $\sigma_W \in S_n$.

- $F_W(G_{\text{LCQP}}) = F_W(\hat{G}_{\text{LCQP}})$, $\forall\, F_W \in \mathcal{F}_{\text{LCQP}}^W$ implies $\Phi_{\text{sol}}(G_{\text{LCQP}}) = \Phi_{\text{sol}}(\hat{G}_{\text{LCQP}})$.

- $F_W(G_{\text{LCQP}})_j = F_W(G_{\text{LCQP}})_{j'}$, $\forall\, F_W \in \mathcal{F}_{\text{LCQP}}^W$ implies $\Phi_{\text{sol}}(G_{\text{LCQP}})_j = \Phi_{\text{sol}}(G_{\text{LCQP}})_{j'}$.

The optimal solution mapping $\Phi_{\text{sol}} : \Phi_{\text{obj}}^{-1}(\mathbb{R}) \to \mathbb{R}^n$ is measurable, i.e., $\Phi_{\text{sol}}^{-1}(A)$ is Lebesgue measurable for any Borel measurable $A \subset \mathbb{R}^n$, and hence by Lusin's theorem, there exists a compact and permutation-invariant subspace $X \subset \Phi_{\text{obj}}^{-1}(\mathbb{R})$ such that $\mathbb{P}[\Phi_{\text{obj}}^{-1}(\mathbb{R})\backslash X] < \epsilon$ and that $\Phi_{\text{sol}}$ restricted on $X$ is continuous. Therefore, applying the generalized Stone-Weierstrass theorem for equivariant functions (Azizian & Lelarge, 2021, Theorem 22), we know that there exists $F_W \in \mathcal{F}_{\text{LCQP}}^W$ satisfying

$$\sup_{G_{\text{LCQP}} \in X} \|F_W(G_{\text{LCQP}}) - \Phi_{\text{sol}}(G_{\text{LCQP}})\| < \delta.$$

Therefore, it holds that

$$\mathbb{P}\left[\|F_W(G_{\text{LCQP}}) - \Phi_{\text{sol}}(G_{\text{LCQP}})\| > \delta\right] \leq \mathbb{P}\left[\Phi_{\text{obj}}^{-1}(\mathbb{R})\backslash X\right] < \epsilon,$$

which completes the proof. $\square$

# B. Proofs for Section 4.1

The proof of Proposition 4.1 is directly from Chen et al. (2023b) since adding a quadratic term in the objective function of an MILP problem does not change the feasible region. However, Propositions 4.2 and 4.3 are not covered in Chen et al. (2023b) and we present their proofs here.

*Proof of Proposisition 4.2.* As discussed in Section 4.1, we consider the following two examples whose optimal objective values are $\frac{9}{2}$ and 6, respectively.

$$\min_{x \in \mathbb{R}^6}\ \frac{1}{2}\sum_{i=1}^{6} x_i^2 + \sum_{i=1}^{6} x_i,$$

s.t. $x_1 + x_2 \geq 1$, $x_2 + x_3 \geq 1$, $x_3 + x_4 \geq 1$,
$x_4 + x_5 \geq 1$, $x_5 + x_6 \geq 1$, $x_6 + x_1 \geq 1$,
$x_j \in \{0, 1\}$, $\forall\, j \in \{1, 2, \ldots, 6\}$.

$$\min_{x \in \mathbb{R}^6}\ \frac{1}{2}\sum_{i=1}^{6} x_i^2 + \sum_{i=1}^{6} x_i,$$

s.t. $x_1 + x_2 \geq 1$, $x_2 + x_3 \geq 1$, $x_3 + x_1 \geq 1$,
$x_4 + x_5 \geq 1$, $x_5 + x_6 \geq 1$, $x_6 + x_4 \geq 1$,
$x_j \in \{0, 1\}$, $\forall\, j \in \{1, 2, \ldots, 6\}$.

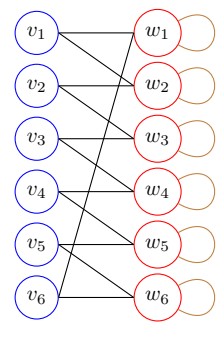
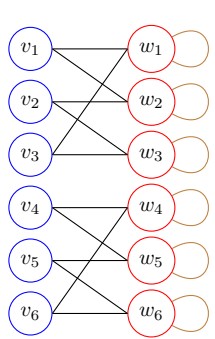

Denote $G_{\text{MI-LCQP}}$ and $\hat{G}_{\text{MI-LCQP}}$ as the graph representations of the above two MI-LCQP problems. Let $s_i^l, t_j^l$ and $\hat{s}_i^l, \hat{t}_j^l$ be the attributes at the $l$-th layer when apply a GNN $F \in \mathcal{F}_{\text{MI-LCQP}}$ to $G_{\text{MI-LCQP}}$ and $\hat{G}_{\text{MI-LCQP}}$. We will prove by induction that for any $0 \le l \le L$, the followings hold:

(a) $s_i^l = \hat{s}_i^l$ and is constant over $i \in \{1, 2, \ldots, 6\}$.

(b) $t_j^l = \hat{t}_j^l$ and is constant over $j \in \{1, 2, \ldots, 6\}$.

It is clear that the conditions (a) and (b) are true for $l = 0$, since $v_i = \hat{v}_i = (1, \ge)$ is constant in $i \in \{1, 2, \ldots, 6\}$, and $w_j = \hat{w}_j = (1, 0, 1, 1)$ is constant in $j \in \{1, 2, \ldots, 6\}$. Now suppose that the conditions (a) and (b) are true for $l - 1$ where $1 \le l \le L$. We denote that $s^{l-1} = s_i^{l-1} = \bar{s}_i^{l-1}, \ \forall \, i \in \{1, 2, \ldots, 6\}$ and $t^{l-1} = t_j^{l-1} = \hat{t}_j^{l-1}, \ \forall \, j \in \{1, 2, \ldots, 6\}$. It can be computed for any $i \in \{1, 2, \ldots, 6\}$ and $j \in \{1, 2, \ldots, 6\}$ that

$$s_i^l = f_l^V \left( s_i^{l-1}, \sum_{j \in \mathcal{N}_i^W} g_l^W(t_j^{l-1}, A_{ij}) \right) = f_l^V \left( s^{l-1}, 2g_l^W(t^{l-1}, 1) \right) = \hat{s}_i^l,$$

$$t_j^l = f_l^W \left( t_j^{l-1}, \sum_{i \in \mathcal{N}_j^V} g_l^V(s_i^{l-1}, A_{ij}), \sum_{j' \in \mathcal{N}_j^W} g_l^Q(t_{j'}^{l-1}, Q_{jj'}) \right)$$

$$= f_l^W \left( t^{l-1}, 2g_l^V(s^{l-1}, 1), g_l^Q(t^{l-1}, 1) \right) = \hat{t}_j^l,$$

which proves (a) and (b) for $l$. Thus, we can conclude that $F(G_{\text{MI-LCQP}}) = F(\hat{G}_{\text{MI-LCQP}}), \ \forall \, F \in \mathcal{F}_{\text{MI-LCQP}}$. □

We also remark that, although the two MI-LCQP graphs with distinct optimal objective values presented above are indistinguishable by GNNs considered in this paper, they can be distinguished by 3-hop GNNs (Feng et al., 2022; Chen et al., 2025). Crucially, when we scale these examples to 10 variables and 10 constraints (one graph with 20 connected nodes and the other with two 10-node components), 3-hop GNNs fail while 5-hop GNNs succeed, confirming that larger $k$ in $k$-hop GNNs reduces GNN-unfriendly cases, though they do not solve all MI-LCQPs.

*Proof of Proposition 4.3.* Consider the following two MI-LCQPs:

$$\min_{x \in \mathbb{R}^7} \ \frac{1}{2} x^\top \mathbf{11}^\top x + \mathbf{1}^\top x,$$

s.t. $x_1 - x_2 = 0, \ x_2 - x_1 = 0,$
$x_3 - x_4 = 0, \ x_4 - x_5 = 0,$
$x_5 - x_6 = 0, \ x_6 - x_7 = 0, \ x_7 - x_3 = 0,$
$x_1 + x_2 + x_3 + x_4 + x_5 + x_6 + x_7 = 6$
$0 \le x_j \le 3, \ x_j \in \mathbb{Z}, \ \forall \, j \in \{1, 2, \ldots, 7\}.$

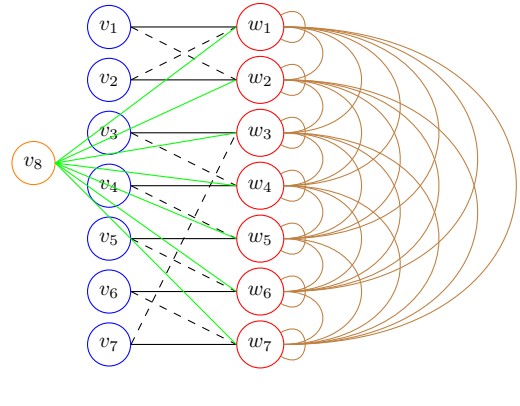

and

$$\min_{x \in \mathbb{R}^7} \ \frac{1}{2} x^\top \mathbf{11}^\top x + \mathbf{1}^\top x,$$

s.t. $x_1 - x_2 = 0, \ x_2 - x_3 = 0, \ x_3 - x_1 = 0,$
$x_4 - x_5 = 0, \ x_5 - x_6 = 0,$
$x_6 - x_7 = 0, \ x_7 - x_4 = 0,$
$x_1 + x_2 + x_3 + x_4 + x_5 + x_6 + x_7 = 6$
$0 \le x_j \le 3, \ x_j \in \mathbb{Z}, \ \forall \, j \in \{1, 2, \ldots, 7\}.$

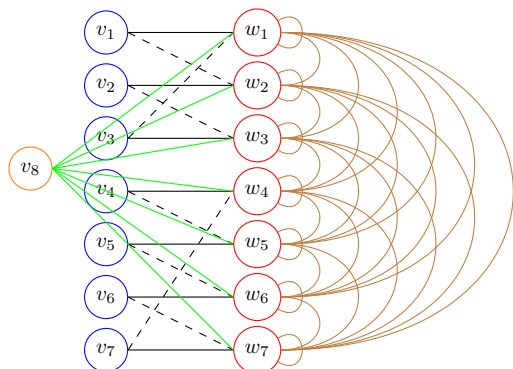

As we mentioned in Section 4.1, both problems are feasible with the same optimal objective value, but have disjoint optimal solution sets.

On the other hand, it can be analyzed using the same argument as in the proof of Proposition 4.2 that for any $0 \leq l \leq L$ that

(a) $s_i^l = \hat{s}_i^l$ is constant over $i \in \{1, 2, \ldots, 7\}$, and $s_8^l = \hat{s}_8^l$.

(b) $t_j^l = \hat{t}_j^l$ is constant over $j \in \{1, 2, \ldots, 7\}$.

These two conditions guarantee that $F(G_{\text{MI-LCQP}}) = F(\hat{G}_{\text{MI-LCQP}})$, $\forall F \in \mathcal{F}_{\text{MI-LCQP}}$ and $F_W(G_{\text{MI-LCQP}}) = F_W(\hat{G}_{\text{MI-LCQP}})$, $\forall F_W \in \mathcal{F}_{\text{MI-LCQP}}$. $\qquad\square$

## C. Proofs for Section 4.2

This section collects the proofs of Theorems 4.6 and 4.7. Similar to the LCQP case, the proofs are also based on the WL test (Algorithm 1) and its separation power to distinguish MI-LCQP problems with different properties. We define the separation power of Algorithm 1 as follows.

**Definition C.1.** Let $G_{\text{MI-LCQP}}, \hat{G}_{\text{MI-LCQP}} \in \mathcal{G}_{\text{MI-LCQP}}^{m,n}$ be two MI-LCQP-graphs and let $\{\!\{C_i^{L,V}\}\!\}_{i=0}^m, \{\!\{C_j^{L,W}\}\!\}_{j=0}^n$ and $\{\!\{\hat{C}_i^{L,V}\}\!\}_{i=0}^m, \{\!\{\hat{C}_j^{L,W}\}\!\}_{j=0}^n$ be color multisets output by Algorithm 1 on $G_{\text{MI-LCQP}}$ and $\hat{G}_{\text{MI-LCQP}}$.

1. We say $G_{\text{MI-LCQP}} \sim \hat{G}_{\text{MI-LCQP}}$ if $\{\!\{C_i^{L,V}\}\!\}_{i=0}^m = \{\!\{\hat{C}_i^{L,V}\}\!\}_{i=0}^m$ and $\{\!\{C_j^{L,W}\}\!\}_{j=0}^n = \{\!\{\hat{C}_j^{L,W}\}\!\}_{j=0}^n$ hold for all $L \in \mathbb{N}$ and all hash functions.

2. We say $G_{\text{MI-LCQP}} \overset{W}{\sim} \hat{G}_{\text{MI-LCQP}}$ if $\{\!\{C_i^{L,V}\}\!\}_{i=0}^m = \{\!\{\hat{C}_i^{L,V}\}\!\}_{i=0}^m$ and $C_j^{L,W} = \hat{C}_j^{L,W}$, $\forall j \in \{1, 2, \ldots, n\}$, for all $L \in \mathbb{N}$ and all hash functions.

The key component in the proof is to show that for GNN-solvable/GNN-analyzable MI-LCQP problems, if they are indistinguishable by the WL test, then they must share some common properties.

**Theorem C.2.** For two GNN-analyzable MI-LCQP-graphs $G_{\text{MI-LCQP}}, \hat{G}_{\text{MI-LCQP}} \in \mathcal{G}_{\text{analyzable}}^{m,n}$, if $G_{\text{MI-LCQP}} \sim \hat{G}_{\text{MI-LCQP}}$, then $\Phi_{\text{feas}}(G_{\text{MI-LCQP}}) = \Phi_{\text{feas}}(\hat{G}_{\text{MI-LCQP}})$ and $\Phi_{\text{obj}}(G_{\text{MI-LCQP}}) = \Phi_{\text{obj}}(\hat{G}_{\text{MI-LCQP}})$.

*Proof.* Let $G_{\text{MI-LCQP}}$ and $\hat{G}_{\text{MI-LCQP}}$ be the MI-LCQP-graphs associated to

$$\min_{x \in \mathbb{R}^n} \frac{1}{2} x^\top Q x + c^\top x, \quad \text{s.t. } Ax \circ b, \, l \leq x \leq u, \, x_j \in \mathbb{Z}, \, \forall j \in I. \tag{15}$$

and

$$\min_{x \in \mathbb{R}^n} \frac{1}{2} x^\top \hat{Q} x + \hat{c}^\top x, \quad \text{s.t. } \hat{A}x \hat{\circ} \hat{b}, \, \hat{l} \leq x \leq \hat{u}, \, x_j \in \mathbb{Z}, \, \forall j \in \hat{I}. \tag{16}$$

Suppose that there are no collisions of hash functions or their linear combinations when applying the WL test to $G_{\text{MI-LCQP}}$ and $\hat{G}_{\text{MI-LCQP}}$ and there are no strict color refinements in the $L$-th iteration. Since $G_{\text{MI-LCQP}} \sim \hat{G}_{\text{MI-LCQP}}$ and both of them are GNN-analyzable, after performing some permutation, there exist $\mathcal{I} = \{I_1, I_2, \ldots, I_s\}$ and $\mathcal{J} = \{J_1, J_2, \ldots, J_t\}$ that are partitions of $\{1, 2, \ldots, m\}$ and $\{1, 2, \ldots, n\}$, respectively, such that the followings hold:

- $C_i^{L,V} = C_{i'}^{L,V}$ if and only if $i, i' \in I_p$ for some $p \in \{1, 2, \ldots, s\}$.

- $C_i^{L,V} = \hat{C}_{i'}^{L,V}$ if and only if $i, i' \in I_p$ for some $p \in \{1, 2, \ldots, s\}$.

- $\hat{C}_i^{L,V} = \hat{C}_{i'}^{L,V}$ if and only if $i, i' \in I_p$ for some $p \in \{1, 2, \ldots, s\}$.

- $C_j^{L,W} = C_{j'}^{L,W}$ if and only if $j, j' \in J_q$ for some $q \in \{1, 2, \ldots, t\}$.

- $C_j^{L,W} = \hat{C}_{j'}^{L,W}$ if and only if $j, j' \in J_q$ for some $q \in \{1, 2, \ldots, t\}$.

- $\hat{C}_j^{L,W} = \hat{C}_{j'}^{L,W}$ if and only if $j, j' \in J_q$ for some $q \in \{1, 2, \ldots, t\}$.

By similar analysis as in the proof of Theorem A.2, we have

(a) $v_i = \hat{v}_i$ and is constant over $i \in I_p$ for any $p \in \{1, 2, \ldots, s\}$.

(b) $w_j = \hat{w}_j$ and is constant over $j \in J_q$ for any $q \in \{1, 2, \ldots, t\}$.

(c) For any $p \in \{1, 2, \ldots, s\}$ and any $q \in \{1, 2, \ldots, t\}$, $\{\!\{A_{ij} : j \in J_q\}\!\} = \{\!\{\hat{A}_{ij} : j \in J_q\}\!\}$ and is constant over $i \in I_p$.

(d) For any $p \in \{1, 2, \ldots, s\}$ and any $q \in \{1, 2, \ldots, t\}$, $\{\!\{A_{ij} : i \in I_p\}\!\} = \{\!\{\hat{A}_{ij} : i \in I_p\}\!\}$ and is constant over $j \in J_q$.

(e) For any $q, q' \in \{1, 2, \ldots, t\}$, $\{\!\{Q_{jj'} : j' \in J_{q'}\}\!\} = \{\!\{\hat{Q}_{jj'} : j' \in J_{q'}\}\!\}$ and is constant over $j \in J_q$.

Note that $G_{\text{MI-LCQP}}$ and $\hat{G}_{\text{MI-LCQP}}$ are both GNN-analyzable, i.e., all submatrices $(A_{ij})_{i \in I_p, j \in J_q}$, $(\hat{A}_{ij})_{i \in I_p, j \in J_q}$, $(Q_{jj'})_{j \in J_q, j' \in J_{q'}}$, and $(\hat{Q}_{jj'})_{j \in J_q, j' \in J_{q'}}$ have identical entries. The above conditions (c)-(e) suggest that

(f) For any $p \in \{1, 2, \ldots, s\}$ and any $q \in \{1, 2, \ldots, t\}$, $A_{ij} = \hat{A}_{ij}$ and is constant over $i \in I_p, j \in J_q$.

(g) For any $q, q' \in \{1, 2, \ldots, t\}$, $Q_{jj'} = \hat{Q}_{jj'}$ and is constant over $j \in J_q, j' \in J_{q'}$.

Combining conditions (a), (b), (f), and (g), we can conclude that $G_{\text{MI-LCQP}}$ and $\hat{G}_{\text{MI-LCQP}}$ are actually identical after applying some permutation, i.e., they are isomorphic, which implies $\Phi_{\text{feas}}(G_{\text{MI-LCQP}}) = \Phi_{\text{feas}}(\hat{G}_{\text{MI-LCQP}})$ and $\Phi_{\text{obj}}(G_{\text{MI-LCQP}}) = \Phi_{\text{obj}}(\hat{G}_{\text{MI-LCQP}})$. $\square$

**MI-LCQP optimal solution mapping $\Phi_{\text{sol}}$.** Before stating the next result, we present the definition of the MI-LCQP optimal solution mapping $\Phi_{\text{sol}}$. Different from the LCQP setting, the optimal solution to an MI-LCQP problem may not exist even if it is feasible and bounded, i.e., $\Phi_{\text{obj}}(G_{\text{MI-LCQP}}) \in \mathbb{R}$. Thus, we have to work with $\mathcal{G}_{\text{sol}}^{m,n} \subset \Phi_{\text{obj}}^{-1}(\mathbb{R}) \subset \mathcal{G}_{\text{MI-LCQP}}^{m,n}$ where $\mathcal{G}_{\text{sol}}^{m,n}$ is the collection of all MI-LCQP-graphs for which an optimal solution exists. For $G_{\text{MI-LCQP}} \in \mathcal{G}_{\text{sol}}^{m,n}$, it is possible that it admits multiple optimal solution. Moreover, there may even exist multiple optimal solutions with the smallest $\ell_2$-norm due to its non-convexity, which means that we cannot define the optimal solution mapping $\Phi_{\text{sol}}$ using the same approach as in the LCQP case. If we further assume that $G_{\text{MI-LCQP}} \in \mathcal{G}_{\text{sol}}^{m,n}$ is GNN-solvable, then using the same approach as in Chen et al. (2023b, Appendix C), one can define a total ordering on the optimal solution set and hence define $\Phi_{\text{sol}}(G_{\text{MI-LCQP}})$ as the minimal element in the optimal solution set, which is unique and permutation-equivariant, meaning that if one relabels vertices of $G_{\text{MI-LCQP}}$, then entries of $\Phi_{\text{sol}}(G_{\text{MI-LCQP}})$ are relabeled accordingly. In particular, since the WL test applied on $G_{\text{MI-LCQP}} \in \mathcal{G}_{\text{sol}}^{m,n} \cap \mathcal{G}_{\text{solvable}}^{m,n}$ yields distinct vertex colors, one can uniquely find a permutation $\sigma \in S_n$ that orders $1, 2, \ldots, n$ lexicographically. Then $\Phi_{\text{sol}}(G_{\text{MI-LCQP}})$ is defined as the vector $x \in \mathbb{R}^n$ so that $(x_{\sigma(1)}, x_{\sigma(2)}, \ldots, x_{\sigma(n)})$ is minimized in the lexicographic sense among all optimal solutions of $G_{\text{MI-LCQP}}$.

**Theorem C.3.** *For any two MI-LCQP-graphs $G_{\text{MI-LCQP}}, \hat{G}_{\text{MI-LCQP}} \in \mathcal{G}_{\text{sol}}^{m,n} \cap \mathcal{G}_{\text{solvable}}^{m,n}$ that are GNN-solvable with nonempty optimal solution sets, if $G_{\text{MI-LCQP}} \sim \hat{G}_{\text{MI-LCQP}}$, then there exists some permutation $\sigma_W \in S_n$ such that $\Phi_{\text{sol}}(G_{\text{MI-LCQP}}) = \sigma_W(\Phi_{\text{sol}}(\hat{G}_{\text{MI-LCQP}}))$. Furthermore, if $G_{\text{MI-LCQP}} \overset{W}{\sim} \hat{G}_{\text{MI-LCQP}}$, then $\Phi_{\text{sol}}(G_{\text{MI-LCQP}}) = \Phi_{\text{sol}}(\hat{G}_{\text{MI-LCQP}})$.*

*Proof.* By Proposition D.1, $G_{\text{MI-LCQP}}$ and $\hat{G}_{\text{MI-LCQP}}$ are also GNN-analyzable, and hence. If $G_{\text{MI-LCQP}} \sim \hat{G}_{\text{MI-LCQP}}$, then they are isomorphic by the analysis in the proof of Theorem C.2, and hence, $\Phi_{\text{sol}}(G_{\text{MI-LCQP}}) = \sigma_W(\Phi_{\text{sol}}(\hat{G}_{\text{MI-LCQP}}))$ for some permutation $\sigma_W \in S_n$. If $G_{\text{MI-LCQP}} \overset{W}{\sim} \hat{G}_{\text{MI-LCQP}}$, then the same analysis in the proof of Theorem C.2 applies and these two graphs are identical after applying some permutation on $V$ with the labeling in $W$ unchanged, which guarantees $\Phi_{\text{sol}}(G_{\text{MI-LCQP}}) = \Phi_{\text{sol}}(\hat{G}_{\text{MI-LCQP}})$. $\square$

As discussed in the main test before Definition 4.4, for GNN-analyzable and GNN-solvable MI-LCQP instances, vertices with essentially different properties will be distinguished by WL test or GNNs. For such GNN-friendly instances, GNNs have provably strong expressive power. In fact, with Theorem C.2 and Theorem C.3, one can prove Theorem 4.6 and Theorem 4.7, via a similar argument when we prove Theorems 3.3 and 3.4.

*Proof of Theorem 4.6.* Theorem 4.6 can be proved based on Theorem C.2. We only present the proof of (7) since (8) and (9) can be proved with almost the same lines.

The separation power of GNNs is equivalent to that of the WL test, i.e., for any $G_{\text{MI-LCQP}}, \hat{G}_{\text{MI-LCQP}} \in \mathcal{G}_{\text{MI-LCQP}}^{m,n}$,

$$G_{\text{MI-LCQP}} \sim \hat{G}_{\text{MI-LCQP}} \iff F(G_{\text{MI-LCQP}}) = F(\hat{G}_{\text{MI-LCQP}}), \ \forall \, F \in \mathcal{F}_{\text{MI-LCQP}}, \tag{17}$$

which combined with Theorem C.2 leads to that for $G_{\text{MI-LCQP}}, \hat{G}_{\text{MI-LCQP}} \in \mathcal{G}_{\text{analyzable}}^{m,n}$,

$$F(G_{\text{MI-LCQP}}) = F(\hat{G}_{\text{MI-LCQP}}), \ \forall \, F \in \mathcal{F}_{\text{MI-LCQP}} \implies \Phi_{\text{feas}}(G_{\text{MI-LCQP}}) = \Phi_{\text{feas}}(\hat{G}_{\text{MI-LCQP}}), \tag{18}$$

indicating that the separation power of $\mathcal{F}_{\text{MI-LCQP}}$ is upper bounded by that of $\Phi_{\text{feas}}$ on $\mathcal{G}_{\text{analyzable}}^{m,n}$.

The function $\Phi_{\text{feas}} : \mathcal{G}_{\text{analyzable}}^{m,n} \to \{0,1\} \subset \mathbb{R}$ is measurable, i.e., $\Phi_{\text{feas}}^{-1}(0)$ and $\Phi_{\text{feas}}^{-1}(0)$ are both Lebesgue measurable, and hence by Lusin's theorem, there exists a compact and permutation-invariant subspace $X \subset \mathcal{G}_{\text{analyzable}}^{m,n}$ such that $\mathbb{P}[\mathcal{G}_{\text{analyzable}}^{m,n} \backslash X] < \epsilon$ and that $\Phi_{\text{feas}}$ restricted on $X$ is continuous. Therefore, by the Stone-Weierstrass theorem and (18), we have that there exists $F \in \mathcal{F}_{\text{MI-LCQP}}$ satisfying

$$\sup_{G_{\text{MI-LCQP}} \in X} |F(G_{\text{MI-LCQP}}) - \Phi_{\text{feas}}(G_{\text{MI-LCQP}})| < \frac{1}{2}$$

Therefore, it holds that

$$\mathbb{P}\left[ \mathbb{I}_{F(G_{\text{MI-LCQP}}) > \frac{1}{2}} \neq \Phi_{\text{feas}}(G_{\text{MI-LCQP}}) \right] \leq \mathbb{P}\left[ \mathcal{G}_{\text{analyzable}}^{m,n} \backslash X \right] < \epsilon,$$

which proves (7). $\qquad\square$

*Proof of Theorem 4.7.* In addition to (17), it can be proved that the separation powers of GNNs and the WL test are equivalent in the following sense:

- For any $G_{\text{MI-LCQP}}, \hat{G}_{\text{MI-LCQP}} \in \mathcal{G}_{\text{MI-LCQP}}^{m,n}$, $G_{\text{MI-LCQP}} \overset{W}{\sim} \hat{G}_{\text{MI-LCQP}}$ if and only if $F_W(G_{\text{MI-LCQP}}) = F_W(\hat{G}_{\text{MI-LCQP}})$ for all $F_W \in \mathcal{F}_{\text{MI-LCQP}}^W$.

- For any $G_{\text{MI-LCQP}} \in \mathcal{G}_{\text{MI-LCQP}}^{m,n}$ and any $j, j' \in W$, $C_j^{L,W} = C_{j'}^{L,W}$ for any $L \in \mathbb{N}$ and any hash function if and only if $F_W(G_{\text{MI-LCQP}})_j = F_W(G_{\text{MI-LCQP}})_{j'}$ for all $F_W \in \mathcal{F}_{\text{MI-LCQP}}^W$.

Therefore, with Theorem C.3, the separation power of GNNs is upper bounded by that of $\Phi_{\text{sol}}$ on $\mathcal{G}_{\text{sol}}^{m,n} \cap \mathcal{G}_{\text{solvable}}^{m,n}$ in the following sense: for any $G_{\text{MI-LCQP}}, \hat{G}_{\text{MI-LCQP}} \in \mathcal{G}_{\text{sol}}^{m,n} \cap \mathcal{G}_{\text{solvable}}^{m,n}$,

- $F(G_{\text{MI-LCQP}}) = F(\hat{G}_{\text{MI-LCQP}}), \ \forall \, F \in \mathcal{F}_{\text{MI-LCQP}}$ implies $\Phi_{\text{sol}}(G_{\text{MI-LCQP}}) = \sigma_W(\Phi_{\text{sol}}(\hat{G}_{\text{MI-LCQP}}))$ for some $\sigma_W \in S_n$.

- $F_W(G_{\text{MI-LCQP}}) = F_W(\hat{G}_{\text{MI-LCQP}}), \ \forall \, F_W \in \mathcal{F}_{\text{MI-LCQP}}^W$ implies $\Phi_{\text{sol}}(G_{\text{MI-LCQP}}) = \Phi_{\text{sol}}(\hat{G}_{\text{MI-LCQP}})$.

- $F_W(G_{\text{MI-LCQP}})_j = F_W(G_{\text{MI-LCQP}})_{j'}, \ \forall \, F_W \in \mathcal{F}_{\text{LCQP}}^W$ implies $j = j'$ and hence $\Phi_{\text{sol}}(G_{\text{MI-LCQP}})_j = \Phi_{\text{sol}}(G_{\text{MI-LCQP}})_{j'}$.

The optimal solution mapping $\Phi_{\text{sol}} : \mathcal{G}_{\text{sol}}^{m,n} \cap \mathcal{G}_{\text{solvable}}^{m,n} \to \mathbb{R}^n$ is measurable, i.e., $\Phi_{\text{sol}}^{-1}(A)$ is Lebesgue measurable for any Borel measurable $A \subset \mathbb{R}^n$, and hence by Lusin's theorem, there exists a compact and permutation-invariant subspace $X \subset \mathcal{G}_{\text{sol}}^{m,n} \cap \mathcal{G}_{\text{solvable}}^{m,n}$ such that $\mathbb{P}[\mathcal{G}_{\text{sol}}^{m,n} \cap \mathcal{G}_{\text{solvable}}^{m,n} \backslash X] < \epsilon$ and that $\Phi_{\text{sol}}$ restricted on $X$ is continuous. Therefore, applying the generalized Stone-Weierstrass theorem for equivariant functions (Azizian & Lelarge, 2021, Theorem 22), we know that there exists $F_W \in \mathcal{F}_{\text{MI-LCQP}}^W$ satisfying

$$\sup_{G_{\text{MI-LCQP}} \in X} \|F_W(G_{\text{MI-LCQP}}) - \Phi_{\text{sol}}(G_{\text{MI-LCQP}})\| < \delta.$$

Therefore, it holds that

$$\mathbb{P}\left[ \|F_W(G_{\text{MI-LCQP}}) - \Phi_{\text{sol}}(G_{\text{MI-LCQP}})\| > \delta \right] \leq \mathbb{P}\left[ \mathcal{G}_{\text{sol}}^{m,n} \cap \mathcal{G}_{\text{solvable}}^{m,n} \backslash X \right] < \epsilon,$$

which completes the proof. $\qquad\square$

**Discussions on various GNN architectures:** In our work we use the sum aggregation, and all results are still valid for the weighted average aggregation. In particular, all our proofs (such as the proof of Theorem A.2) hold almost verbatimly for the average aggregation. The attention aggregation (Veličković et al., 2017) has stronger separation power, which implies that all universal approximation results still hold. Moreover, all the counter examples for MI-LCQPs work for every aggregation approach, since the color refinement in Algorithm 1 is implemented on multisets, with separation power stronger than or equal to all aggregations of neighboring information.

## D. Characterization of GNN-analyzability and GNN-solvability

This section provides a detailed discussion of these conditions for MI-LCQP graphs, as defined in Section 4.3.

### D.1. Relationship between GNN-analyzability and GNN-solvability

We first prove that GNN-solvability implies GNN-analyzability but they are not equivalent.

**Proposition D.1.** *If $G_{\text{MI-LCQP}} \in \mathcal{G}_{\text{MI-LCQP}}^{m,n}$ is GNN-solvable, then it is also GNN-analyzable.*

*Proof.* Let $(\mathcal{I}, \mathcal{J})$ be the final stable partition of $V \cup W$ generated by WL test on $G_{\text{MI-LCQP}}$ without collision, where $\mathcal{I} = \{I_1, I_2, \ldots, I_s\}$ is a partition of $V = \{1, 2, \ldots, m\}$ and $\mathcal{J} = \{J_1, J_2, \ldots, J_t\}$ is a partition of $W = \{1, 2, \ldots, n\}$. Since we assume that $G_{\text{MI-LCQP}}$ is GNN-insolvable, we have $t = n$ and $|J_1| = |J_2| = \cdots = |J_n| = 1$. Then for any $q, q' \in \{1, 2, \ldots, t\}$, the submatrix $(Q_{jj'})_{j \in J_q, j' \in J_{q'}}$ is a $1 \times 1$ matrix and hence has identical entries.

Consider any $p \in \{1, 2, \ldots, s\}$ and $q \in \{1, 2, \ldots, t\}$. Suppose that the color positioning is stabilized at the $L$-th iteration of WL test. Then for any $i, i' \in I_p$, we have

$$C_i^{L,V} = C_{i'}^{L,V}$$
$$\implies \left\{\!\!\left\{ \text{hash}\left( C_j^{L-1,W}, A_{ij} \right) : j \in \mathcal{N}_i^W \right\}\!\!\right\} = \left\{\!\!\left\{ \text{hash}\left( C_j^{L-1,W}, A_{i'j} \right) : j \in \mathcal{N}_i^W \right\}\!\!\right\}$$
$$\implies \{\!\{A_{ij} : j \in J_q\}\!\} = \{\!\{A_{i'j} : j \in J_q\}\!\},$$

which implies that the submatrix $(A_{ij})_{i \in I_p, j \in J_q}$ has identical entries since $|J_q| = 1$. Therefore, $G_{\text{MI-LCQP}}$ is GNN-analyzable. $\square$

**Proposition D.2.** *There exist GNN-analyzable instances in $\mathcal{G}_{\text{MI-LCQP}}^{m,n}$ that are not GNN-solvable.*

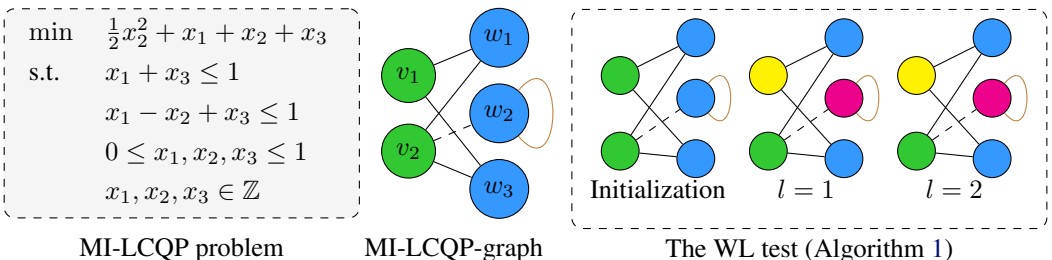

MI-LCQP problem      MI-LCQP-graph      The WL test (Algorithm 1)

*Figure 6.* Example for proving Proposition D.2

*Proof.* Consider the example in Figure 6, for which the final stable partition is $\mathcal{I} = \{\!\{1\}, \{2\}\!\}$ and $\mathcal{J} = \{\!\{1, 3\}, \{2\}\!\}$. It is not GNN-solvable since the class $\{1, 3\}$ in $\mathcal{J}$ has two elements. However, it is GNN-analyzable since $A_{11} = A_{13} = 1$ and $A_{21} = A_{23} = 1$. $\square$

### D.2. Frequency of GNN-analyzability and GNN-solvability

It can be proved that a generic MI-LCQP-graph in $\mathcal{G}_{\text{MI-LCQP}}^{m,n}$ is GNN-solvable almost surely under some mild conditions. Intuitively, if $c \in \mathbb{R}^n$ is randomly sampled from a continuous distribution with density, then almost surely it holds that $x_j \neq x_{j'}$ for any $j \neq j'$, which implies that the vertices in $W$ have different colors initially and always, if there are no collisions of hash functions.

**Proposition D.3.** *Let $\mathbb{P}$ be a probability measure over $\mathcal{G}_{\text{MI-LCQP}}$ such that the marginal distribution $\mathbb{P}_c$ of $c \in \mathbb{R}^n$ has density. Then $\mathbb{P}[\mathcal{G}_{\text{MI-LCQP}} \in \mathcal{G}_{\text{solvable}}^{m,n}] = 1$.*

*Proof.* Since the marginal distribution $\mathbb{P}_c$ has density, almost surely we have for any $j \neq j'$ that

$$c_j \neq c_{j'} \implies C_j^{0,W} \neq C_{j'}^{0,W} \implies C_j^{l,W} \neq C_{j'}^{l,W}, \quad \forall\, l \geq 0,$$

where we assumed that no collisions happen in hash functions. Therefore, any $j, j' \in W$ with $j \neq j'$ are not the in same class of the final stable partition $(\mathcal{I}, \mathcal{J})$, which proves the GNN-solvability. $\qquad\square$

As a direct corollary of Proposition D.1 and Proposition D.3, a generic MI-LCQP-graph in $\mathcal{G}_{\text{MI-LCQP}}^{m,n}$ must also be GNN-analyzable.

**Corollary D.4.** *Let $\mathbb{P}$ be a probability measure over $\mathcal{G}_{\text{MI-LCQP}}$ such that the marginal distribution $\mathbb{P}_c$ of $c \in \mathbb{R}^n$ has density. Then $\mathbb{P}[\mathcal{G}_{\text{MI-LCQP}} \in \mathcal{G}_{\text{analyzable}}^{m,n}] = 1$.*

**Frequency of GNN-friendly instances in practice.** While MI-LCQPs are almost surely GNN-solvable under some assumption, however, in practice, particularly with manually designed instances (as opposed to randomly generated ones), hard instances (GNN-insolvable, or even inanalyzable) does occur. We performed experiments on QPLIB (https://qplib.zib.de/). In this dataset, 19 out of 96 binary-variable linear constraint instances are GNN-insolvable. Furthermore, when coefficients are quantized (which increases the level of symmetry in the dataset), the ratio of GNN-insolvable instances increases. (shown in the table below)

Quantization refers to rounding continuous or high-precision values to discrete levels. For example, rounding coefficients of QP instances to the nearest integer reduces precision but can simplify analysis and potentially uncover additional properties. In our study, we examine instances at different quantization step sizes (0.1, 0.5, and 1).

While GNN-insolvable instances are generally challenging for GNNs, a significant proportion of these hard instances are GNN-analyzable. For example, with a quantization step size of 1 (rounding coefficients to integers), 36 out of 63 GNN-insolvable instances are GNN-analyzable. According to our theorems, GNNs can at least predict feasibility and boundedness (objective value) for such instances. Detailed results are as follows:

*Table 2.* Frequency of GNN-analyzability and solvability

|  | Original data set | Step size 0.1 | Step size 0.5 | Step size 1 |
|---|---|---|---|---|
| Total | 96 | 96 | 96 | 96 |
| GNN-insolvable | 19 | 23 | 42 | 63 |
| GNN-insolvable but analyzable | 0 | 1 | 19 | 36 |

**How to handle bad instances?** If a dataset contains a significant proportion of GNN-insolvable or GNN-inanalyzable instances (highly symmetric structures), we suggest two potential approaches: (I) Adding features: Introducing additional features can differentiate nodes in symmetric graphs. For example, adding a random feature to nodes with identical attributes ensures they are no longer symmetric (Sato et al., 2021). (II) Using higher-order GNNs: These models can distinguish nodes that standard message-passing GNNs cannot, enhancing their expressive power (Morris et al., 2019). In particular, we highlight the potential of $k$-hop GNNs (Feng et al., 2022), which have greater expressive power than message-passing GNNs (MP-GNNs). In the proof of Proposition 4.2 (Appendix B), we construct MI-LCQP instances with different optimal objective values that MP-GNNs cannot distinguish. However, 3-hop GNNs are able to distinguish them. When these instances are scaled to graphs with 10 variables and 10 constraints (resulting in one 20-node connected graph vs. two disjoint 10-node graphs), 3-hop GNNs fail while 5-hop GNNs succeed. This observation is investigated in (Chen et al., 2025) on general graphs and suggests that increasing $k$ reduces the number of indistinguishable cases and may enhance solvability.

## E. Extension to quadratically constrained quadratic programs

A general quadratically constrained quadratic programming (QCQP) is given by

$$\min_{x \in \mathbb{R}^n} \ \frac{1}{2} x^\top Q x + c^\top x, \quad \text{s.t.} \ \frac{1}{2} x^\top P_i x + a_i^\top x \leq b_i, \ 1 \leq i \leq m, \ l \leq x \leq u, \tag{19}$$

where $Q, P_i \in \mathbb{R}^{n \times n}$ are symmetric, $c, a_i \in \mathbb{R}^n$, $b_i \in \mathbb{R}$, $l \in (\mathbb{R} \cup \{-\infty\})^n$, and $u \in (\mathbb{R} \cup \{+\infty\})^n$. We denote $A = \begin{bmatrix} a_1 & a_2 & \cdots & a_m \end{bmatrix}^\top \in \mathbb{R}^{m \times n}$ for consistent notation with (2).

## E.1. Graph representation and GNNs for QCQPs

**Graph representation for QCQPs**   The QCQP-graph for representing (19) is based on the LCQP-graph introduced in Section 2. More specifically, The QCQP graph can be constructed by incorporating the information from $P = (P_1, P_2, \ldots, P_m)$ into the LCQP graph:

- The multiset $\{\!\{ i, j, j' \}\!\}$ is viewed as a hyperedge with weight $(H_i)_{jj'}$ for each $i \in V$ and $j, j' \in W$, where $j = j'$ is allowed.

We use $\mathcal{G}_{\mathrm{QCQP}}^{m,n}$ to denote the set of all QCQP-graphs with $m$ constraints and $n$ variables.

**GNNs for solving QCQP**   Note GNNs on LCQP-graphs that iterate vertex features with message-passing mechanism, which does not naturally adapt to the hyperedges in QCQP graphs. Thus, one idea is to add edge features for each pair $(i, j)$, $i \in V, j \in W$. We describe the GNN architecture for QCQP tasks in detail as follows.

The initial layer computes node features $s_i^0, t_j^0$ and edge features $e_{ij}^0$ via embedding:

- $s_i^0 = f_0^V(v_i)$ for $i \in V$,
- $t_j^0 = f_0^W(w_j)$ for $j \in W$, and
- $e_{ij}^0 = f_0^E(A_{ij})$ for $i \in V, j \in W$.

The $l$-th message-passing layers ($l = 1, 2, \ldots, L$) update the node features using neighbors' information:

- $s_i^l = f_l^V\big(s_i^{l-1}, \sum_{j \in W} g_l^V(t_j^{l-1}, e_{ij}^{l-1})\big)$ for $i \in V$,
- $t_j^l = f_l^W\big(t_j^{l-1}, \sum_{i \in V} g_l^W(s_i^{l-1}, e_{ij}^{l-1}), \sum_{j' \in W} Q_{jj'} g_l^Q(t_{j'}^{l-1})\big)$ for $j \in W$, and
- $e_{ij}^l = f_l^E\big(e_{ij}^{l-1}, \sum_{j' \in W} (P_i)_{jj'} g_l^E(t_{j'}^{l-1})\big)$ for $i \in V, j \in W$.

Finally, there are two types of output layers. The graph-level output computes a single real number for the whole graph

- $y = r_1\big(\sum_{i \in V} s_i^L, \sum_{j \in W} t_j^L\big) \in \mathbb{R}$,

and the node-level output computes a vector $y \in \mathbb{R}^n$ with the $j$-th entry being

- $y_j = r_2\big(\sum_{i \in V} s_i^L, \sum_{j \in W} t_j^L, t_j^L\big)$.

We use $\mathcal{F}_{\mathrm{QCQP}}$ (or $\mathcal{F}_{\mathrm{QCQP}}^W$) to denote the collection of all message-passing GNNs with graph-level (or node-level) outputs that are constructed by continuous $f_0^V, f_0^W, f_0^E, f_l^V, f_l^W, f_l^E, g_l^V, g_l^W, g_l^E, g_l^Q$ ($1 \le l \le L$), and $r_1$ (or $r_2$).

## E.2. Universal Approximation of GNNs for QCQPs

For QCQPs, we still consider the three target mappings, i.e., the feasible mapping $\Phi_{\mathrm{feas}} : \mathcal{G}_{\mathrm{QCQP}}^{m,n} \to \{0, 1\}$, the optimal objective value mapping $\Phi_{\mathrm{obj}} : \mathcal{G}_{\mathrm{QCQP}}^{m,n} \to \mathbb{R} \cup \{\pm\infty\}$, and the optimal solution mapping $\Phi_{\mathrm{obj}}$ that computes the unique optimal solution with the smallest $\ell_2$-norm of feasible and bounded QCQPs with $Q, P_i \succeq 0$, $i = 1, 2, \ldots, m$. The main results that GNNs can universally approximate these three target mappings are stated as follows.

**Assumption E.1.** $\mathbb{P}$ is a Borel regular probability measure on $\mathcal{G}_{\mathrm{QCQP}}^{m,n}$[6].

---

[6]The space $\mathcal{G}_{\mathrm{QCQP}}^{m,n}$ is equipped with the subspace topology induced from the product space $\big\{(A, b, c, Q, P, l, u, \circ) : A \in \mathbb{R}^{m \times n}, b \in \mathbb{R}^m, c \in \mathbb{R}^n, Q \in \mathbb{R}^{n \times n}, P \in (\mathbb{R}^{n \times n})^m, l \in (\mathbb{R} \cup \{-\infty\})^n, u \in (\mathbb{R} \cup \{+\infty\})^n\big\}$, where all Euclidean spaces have standard Eudlidean topologies, discrete spaces $\{-\infty\}$ and $\{+\infty\}$ have the discrete topologies, and all unions are disjoint unions.

**Theorem E.2.** *Let $\mathbb{P}$ be a probability measure satisfying Assumption E.1 and $\mathbb{P}[Q \succeq 0] = \mathbb{P}[P_i \succeq 0] = 1$, $i = 1, 2, \ldots, m$. For any $\epsilon > 0$, there exists $F \in \mathcal{F}_{\text{MI-LCQP}}$ such that*

$$\mathbb{P}\left[\mathbb{I}_{F(G_{\text{QCQP}}) > \frac{1}{2}} \neq \Phi_{\text{feas}}(G_{\text{QCQP}})\right] < \epsilon.$$

**Theorem E.3.** *Let $\mathbb{P}$ be a probability measure satisfying Assumption E.1 and $\mathbb{P}[Q \succeq 0] = \mathbb{P}[P_i \succeq 0] = 1$, $i = 1, 2, \ldots, m$. For any $\epsilon > 0$, there exists $F_1 \in \mathcal{F}_{\text{QCQP}}$ such that*

$$\mathbb{P}\left[\mathbb{I}_{F_1(G_{\text{QCQP}}) > \frac{1}{2}} \neq \mathbb{I}_{\Phi_{\text{obj}}(G_{\text{QCQP}}) \in \mathbb{R}}\right] < \epsilon.$$

*Additionally, if $\mathbb{P}[\Phi_{\text{obj}}(G_{\text{QCQP}}) \in \mathbb{R}] = 1$, for any $\epsilon, \delta > 0$, there exists $F_2 \in \mathcal{F}_{\text{QCQP}}$ such that*

$$\mathbb{P}\left[|F_2(G_{\text{QCQP}}) - \Phi_{\text{obj}}(G_{\text{QCQP}})| > \delta\right] < \epsilon.$$

**Theorem E.4.** *Let $\mathbb{P}$ be a probability measure satisfying Assumption E.1 and $\mathbb{P}[Q \succeq 0] = \mathbb{P}[P_i \succeq 0] = 1$, $i = 1, 2, \ldots, m$. For any $\epsilon, \delta > 0$, there exists $F_W \in \mathcal{F}_{\text{QCQP}}^W$ such that*

$$\mathbb{P}\left[\|F_W(G_{\text{QCQP}}) - \Phi_{\text{sol}}(G_{\text{QCQP}})\| > \delta\right] < \epsilon.$$

Similarly, the proofs of Theorem E.2, E.3, and E.4 are based on showing that the WL test associated with the GNN classes $\mathcal{F}_{\text{QCQP}}$ and $\mathcal{F}_{\text{QCQP}}^W$ have sufficiently strong separation power to distinguish QCQP problems with different properties. We will present and prove such separation results (Theorem E.5, Theorem E.6, and Corollary E.7) in the rest of this subsection, and do not repeat the same arguments as described in the Proof of Theorem 3.3 and Theorem 3.4.

We state in Algorithm 3 the WL test for QCQPs. For QCQP-graphs $G_{\text{QCQP}}, \hat{G}_{\text{QCQP}} \in \mathcal{G}_{\text{QCQP}}^{m,n}$,

1. We say $G_{\text{QCQP}} \sim \hat{G}_{\text{QCQP}}$ if $\{\!\{C_i^{L,V}\}\!\}_{i=0}^m = \{\!\{\hat{C}_i^{L,V}\}\!\}_{i=0}^m$ and $\{\!\{C_j^{L,W}\}\!\}_{j=0}^n = \{\!\{\hat{C}_j^{L,W}\}\!\}_{j=0}^n$ hold for all $L \in \mathbb{N}$ and all hash functions.

2. We say $G_{\text{QCQP}} \overset{W}{\sim} \hat{G}_{\text{QCQP}}$ if $\{\!\{C_i^{L,V}\}\!\}_{i=0}^m = \{\!\{\hat{C}_i^{L,V}\}\!\}_{i=0}^m$ and $C_j^{L,W} = \hat{C}_j^{L,W}$, $\forall j \in \{1, 2, \ldots, n\}$, for all $L \in \mathbb{N}$ and all hash functions.

---

**Algorithm 3** The WL test for QCQP-Graphs

---

**Require:** A QCQP-graph $G = (V, W, A, Q, P, H_V, H_W)$ and iteration limit $L > 0$.

1: Initialize with
$$C_i^{0,V} = \text{hash}(v_i), \ C_j^{0,W} = \text{hash}(w_j), \ C_{ij}^{0,E} = \text{hash}(A_{ij}).$$

2: **for** $l = 1, 2, \cdots, L$ **do**

3: Refine the color

$$C_i^{l,V} = \text{hash}\left(C_i^{l-1,V}, \sum_{j \in W} \text{hash}\left(C_j^{l-1,W}, C_{ij}^{l-1,E}\right)\right),$$

$$C_j^{l,W} = \text{hash}\left(C_j^{l-1,W}, \sum_{i \in V} \text{hash}\left(C_i^{l-1,V}, C_{ij}^{l-1,E}\right), \sum_{j' \in W} Q_{jj'}\text{hash}(C_{j'}^{l-1,W})\right),$$

$$C_{ij}^{l,E} = \text{hash}\left(C_{ij}^{l-1,E}, \sum_{j' \in W} (P_i)_{jj'}\text{hash}(C_{j'}^{l-1,W})\right).$$

4: **end for**

5: **return** The multisets containing all vertex colors $\{\!\{C_i^{L,V}\}\!\}_{i=0}^m, \{\!\{C_j^{L,W}\}\!\}_{j=0}^n$.

---

**Theorem E.5.** *Given $G_{\text{QCQP}}, \hat{G}_{\text{QCQP}} \in \mathcal{G}_{\text{QCQP}}^{m,n}$ with $Q, \hat{Q}, P_i, \hat{P}_i \succeq 0$ for all $i \in \{1, 2, \ldots, m\}$, if $G_{\text{QCQP}} \sim \hat{G}_{\text{QCQP}}$, then $\Phi_{\text{feas}}(G_{\text{QCQP}}) = \Phi_{\text{feas}}(\hat{G}_{\text{QCQP}})$ and $\Phi_{\text{obj}}(G_{\text{QCQP}}) = \Phi_{\text{obj}}(\hat{G}_{\text{QCQP}})$.*

*Proof.* We only show the proof of $\Phi_{\text{obj}}(G_{\text{QCQP}}) = \Phi_{\text{obj}}(\hat{G}_{\text{QCQP}})$ and $\Phi_{\text{feas}}(G_{\text{QCQP}}) = \Phi_{\text{feas}}(\hat{G}_{\text{QCQP}})$ will be a direct corollary.

Let $G_{\text{QCQP}}$ and $\hat{G}_{\text{QCQP}}$ be the QCQP-graph associated to (19) and

$$\min_{x \in \mathbb{R}^n} \frac{1}{2} x^\top \hat{Q} x + \hat{c}^\top x, \quad \text{s.t.} \quad \frac{1}{2} x^\top \hat{P}_i x + \hat{a}_i^\top x \le \hat{b}_i, \ 1 \le i \le m, \ \hat{l} \le x \le \hat{u}, \tag{20}$$

Suppose that there are no collisions of hash functions or their linear combinations when applying the WL test to $G$ and $\hat{G}$ and there are no strict color refinements in the $L$-th iteration. Since $G$ and $\hat{G}$ are indistinguishable by the WL test, after performing some permutation, there exist $\mathcal{I} = \{I_1, I_2, \ldots, I_s\}$ and $\mathcal{J} = \{J_1, J_2, \ldots, J_t\}$ that are partitions of $\{1, 2, \ldots, m\}$ and $\{1, 2, \ldots, n\}$, respectively, such that the followings hold:

- $C_i^{L,V} = C_{i'}^{L,V}$ if and only if $i, i' \in I_p$ for some $p \in \{1, 2, \ldots, s\}$.

- $C_i^{L,V} = \hat{C}_{i'}^{L,V}$ if and only if $i, i' \in I_p$ for some $p \in \{1, 2, \ldots, s\}$.

- $\hat{C}_i^{L,V} = \hat{C}_{i'}^{L,V}$ if and only if $i, i' \in I_p$ for some $p \in \{1, 2, \ldots, s\}$.

- $C_j^{L,W} = C_{j'}^{L,W}$ if and only if $j, j' \in J_q$ for some $q \in \{1, 2, \ldots, t\}$.

- $C_j^{L,W} = \hat{C}_{j'}^{L,W}$ if and only if $j, j' \in J_q$ for some $q \in \{1, 2, \ldots, t\}$.

- $\hat{C}_j^{L,W} = \hat{C}_{j'}^{L,W}$ if and only if $j, j' \in J_q$ for some $q \in \{1, 2, \ldots, t\}$.

The followings hold by the same arguments as in the proof of Theorem A.2:

- $b_i = \hat{b}_i$ and is constant over $i \in I_p$, for any $p \in \{1, 2, \ldots, s\}$.

- $(c_j, l_j, u_j) = (\hat{c}_j, \hat{l}_j, \hat{u}_j)$ and is constant over $j \in J_q$ for any $q \in \{1, 2, \ldots, t\}$.

- For any $p \in \{1, 2, \ldots, s\}$ and $q \in \{1, 2, \ldots, t\}$, $\sum_{j \in J_q} A_{ij} = \sum_{j \in J_q} \hat{A}_{ij}$ and is constant over $i \in I_p$.

- For any $p \in \{1, 2, \ldots, s\}$ and $q \in \{1, 2, \ldots, t\}$, $\sum_{i \in I_p} A_{ij} = \sum_{i \in I_p} \hat{A}_{ij}$ and is constant over $j \in J_q$.

- For any $q, q' \in \{1, 2, \ldots, t\}$, $\sum_{j' \in J_{q'}} Q_{jj'} = \sum_{j' \in J_{q'}} \hat{Q}_{jj'}$ and is constant over $j \in J_q$.

Fix $p \in \{1, 2, \ldots, s\}$ and $q, q' \in \{1, 2, \ldots, t\}$. For any $j, j' \in J_q$, we have

$$C_j^{L,W} = C_{j'}^{L,W}$$
$$\implies \sum_{i \in V} \text{hash}\left(C_i^{L-1,V}, C_{ij}^{L-1,E}\right) = \sum_{i \in V} \text{hash}\left(C_i^{L-1,V}, C_{ij'}^{L-1,E}\right)$$
$$\implies \left\{\left\{C_{ij}^{L,E} : i \in I_p\right\}\right\} = \left\{\left\{C_{ij'}^{L,E} : i \in I_p\right\}\right\}$$
$$\implies \left\{\left\{\sum_{j'' \in W} (P_i)_{jj''}\text{hash}(C_{j''}^{L-1,W}) : i \in I_p\right\}\right\}$$
$$= \left\{\left\{\sum_{j'' \in W} (P_i)_{j'j''}\text{hash}(C_{j''}^{L-1,W}) : i \in I_p\right\}\right\}$$
$$\implies \left\{\left\{\sum_{j'' \in J_{q'}} (P_i)_{jj''} : i \in I_p\right\}\right\} = \left\{\left\{\sum_{j'' \in J_{q'}} (P_i)_{j'j''} : i \in I_p\right\}\right\}$$
$$\implies \sum_{j'' \in J_{q'}} \sum_{i \in I_p} (P_i)_{jj''} = \sum_{j'' \in J_{q'}} \sum_{i \in I_p} (P_i)_{j'j''}.$$

One can do a similar analysis for $C_j^{L,W} = \hat{C}_{j'}^{L,W}$ and $\hat{C}_j^{L,W} = \hat{C}_{j'}^{L,W}$ where $j, j' \in J_q$. This concludes that

$$\sum_{j' \in J_{q'}} \sum_{i \in I_p} (P_i)_{jj'} = \sum_{j' \in J_{q'}} \sum_{i \in I_p} (\hat{P}_i)_{jj'}$$

is constant over $j \in J_q$.

Let $x \in \mathbb{R}^n$ be any feasible solution to (19) and define $\hat{x} \in \mathbb{R}^n$ via $\hat{x}_j = y_q = \frac{1}{|J_q|} \sum_{j' \in J_q} x_{j'}$ for $j \in J_q$. For any $p \in \{1, 2, \ldots, s\}$, it follows from

$$\frac{1}{2} x^\top P_i x + a_i^\top x \leq b_i, \quad i \in I_p,$$

and Lemma A.4 that

$$\frac{1}{I_p} \sum_{i \in I_p} \hat{b}_i = \frac{1}{I_p} \sum_{i \in I_p} b_i \geq \frac{1}{2} x^\top \left( \frac{1}{|I_p|} \sum_{i \in I_p} P_i \right) x + \left( \frac{1}{I_p} \sum_{i \in I_p} a_i \right)^\top x$$

$$\geq \frac{1}{2} \hat{x}^\top \left( \frac{1}{|I_p|} \sum_{i \in I_p} P_i \right) \hat{x} + \left( \frac{1}{I_p} \sum_{i \in I_p} a_i \right)^\top \hat{x} = \frac{1}{2} \hat{x}^\top \left( \frac{1}{|I_p|} \sum_{i \in I_p} \hat{P}_i \right) \hat{x} + \left( \frac{1}{I_p} \sum_{i \in I_p} \hat{a}_i \right)^\top \hat{x}.$$

Note that for any $i, i' \in I_p$ and any $q, q' \in \{1, 2, \ldots, t\}$, we have

$$\hat{C}_i^{L,V} = \hat{C}_{i'}^{L,V}$$

$$\implies \sum_{j \in W} \text{hash}\left( \hat{C}_j^{L-1,W}, \hat{C}_{ij}^{L-1,E} \right) = \sum_{j \in W} \text{hash}\left( \hat{C}_j^{L-1,W}, \hat{C}_{i'j}^{L-1,E} \right)$$

$$\implies \left\{\left\{ \hat{C}_{ij}^{L,E} : j \in J_q \right\}\right\} = \left\{\left\{ \hat{C}_{i'j}^{L,E} : j \in J_q \right\}\right\}$$

$$\implies \left\{\left\{ \sum_{j' \in W} (\hat{P}_i)_{jj'} \text{hash}(\hat{C}_{j'}^{L-1,W}) : j \in J_q \right\}\right\}$$

$$= \left\{\left\{ \sum_{j' \in W} (\hat{P}_{i'})_{jj'} \text{hash}(\hat{C}_{j'}^{L-1,W}) : j \in J_q \right\}\right\}$$

$$\implies \left\{\left\{ \sum_{j' \in J_{q'}} (\hat{P}_i)_{jj'} : j \in J_q \right\}\right\} = \left\{\left\{ \sum_{j' \in J_{q'}} (\hat{P}_{i'})_{jj'} : j \in J_q \right\}\right\}$$

$$\implies \sum_{j \in J_q} \sum_{j' \in J_{q'}} (\hat{P}_i)_{jj'} = \sum_{j \in J_q} \sum_{j' \in J_{q'}} (\hat{P}_{i'})_{jj'}.$$

Therefore, it holds that

$$\frac{1}{2} \hat{x}^\top \left( \frac{1}{|I_p|} \sum_{i' \in I_p} \hat{P}_{i'} \right) \hat{x} = \frac{1}{2} \hat{x}^\top \hat{P}_i \hat{x}, \quad \forall \, i \in I_p,$$

and hence that

$$\frac{1}{2} \hat{x}^\top P_i \hat{x} + \hat{a}_i^\top x \leq \hat{b}_i, \quad \forall \, i \in I_p.$$

We thus know that $\hat{x}$ is a feasible solution to (12). In addition, we have

$$\frac{1}{2} x^\top Q x + c^\top x \geq \frac{1}{2} \hat{x}^\top Q \hat{x} + c^\top \hat{x} = \frac{1}{2} \hat{x}^\top \hat{Q} \hat{x} + \hat{c}^\top \hat{x},$$

which implies that $\Phi_{\text{obj}}(G_{\text{QCQP}}) \geq \Phi_{\text{obj}}(\hat{G}_{\text{QCQP}})$. The reverse direction $\Phi_{\text{obj}}(G_{\text{QCQP}}) \leq \Phi_{\text{obj}}(\hat{G}_{\text{QCQP}})$ is also true and we can conclude that $\Phi_{\text{obj}}(G_{\text{QCQP}}) = \Phi_{\text{obj}}(\hat{G}_{\text{QCQP}})$. $\qquad \square$

**Theorem E.6.** *For any $G_{\text{QCQP}}, \hat{G}_{\text{QCQP}} \in \mathcal{G}_{\text{QCQP}}^{m,n}$ with $Q, \hat{Q}, P_i, \hat{P}_i \succeq 0$, $i \in \{1, 2, \ldots, m\}$ that are feasible and bounded, if $G_{\text{QCQP}} \sim \hat{G}_{\text{QCQP}}$, then there exists some permutation $\sigma_W \in S_n$ such that $\Phi_{\text{sol}}(G_{\text{QCQP}}) = \sigma_W(\Phi_{\text{sol}}(\hat{G}_{\text{QCQP}}))$. Furthermore, if $G_{\text{QCQP}} \overset{W}{\sim} \hat{G}_{\text{QCQP}}$, then $\Phi_{\text{sol}}(G_{\text{QCQP}}) = \Phi_{\text{sol}}(\hat{G}_{\text{QCQP}})$.*

*Proof.* Based on Theorem E.5, Theorem E.6 can be proved by the same arguments as in the proof of Lemma B.4 and Corollary B.7 in Chen et al. (2023a), which is included in the proof of Theorem A.2. $\square$

**Corollary E.7.** *For any $G_{\text{QCQP}} \in \mathcal{G}_{\text{QCQP}}^{m,n}$ that is feasible and bounded and any $j, j' \in \{1, 2, \ldots, n\}$, if $C_j^{L,W} = C_{j'}^{L,W}$ holds for all $L \in \mathbb{N}_+$ and all hash functions, then $\Phi_{\text{sol}}(G_{\text{QCQP}})_j = \Phi_{\text{sol}}(G_{\text{QCQP}})_{j'}$.*

*Proof.* Let $\hat{G}_{\text{QCQP}}$ be the QCQP-graph obtained from $G_{\text{QCQP}}$ by relabeling $j$ as $j'$ and relabeling $j'$ as $j$. By Theorem E.6, we have $\Phi_{\text{sol}}(G_{\text{QCQP}}) = \Phi_{\text{sol}}(\hat{G}_{\text{QCQP}})$, which implies $\Phi_{\text{sol}}(G_{\text{QCQP}})_j = \Phi_{\text{sol}}(\hat{G}_{\text{QCQP}})_j = \Phi_{\text{sol}}(G_{\text{QCQP}})_{j'}$. $\square$

# F. Potential extension to polynomial optimization

Beyond QCQPs, the hyperedge-based approach may extend to more complex settings such as polynomial optimization. We outline a high-level idea: consider a monomial term $F_{j_1, j_2, \ldots, j_k} x_{j_1} x_{j_2} \cdots x_{j_k}$ in a polynomial objective or constraint. If the term appears in the objective, we model it as a hyperedge $\{\!\!\{ w_{j_1}, w_{j_2}, \ldots, w_{j_k} \}\!\!\}$; if it appears in the $i$-th constraint, we model it as $\{\!\!\{ v_i, w_{j_1}, w_{j_2}, \ldots, w_{j_k} \}\!\!\}$. We conjecture that GNNs operating on such hypergraphs can approximate key properties of convex polynomial optimization. As supporting evidence, our QCQP analysis in Appendix E shows that second-order hypergraph GNNs already achieve universal approximation for core properties, suggesting promise for broader generalizations.

# G. Implementation details and additional numerical results

In this section, we explain how we formulate the optimization problems used in the numerical experiments and how to randomly generate problem instances. We mainly follow the settings of OSQP (Stellato et al., 2020) with slight modifications.

## G.1. Random LCQP and MI-LCQP instance generation

**Generic LCQP and MI-LCQP generation.** For all instances generated and used in our numerical experiments, we set $m = 10$ and $n = 50$, which means each instance contains 10 constraints and 50 variables. The sampling schemes of problem components are described below.

- Matrix $Q$ in the objective function. We sample sparse, symmetric and positive semidefinite $Q$ using the `make_sparse_spd_matrix` function provided by the `scikit-learn` Python package, which imposes sparsity on the Cholesky factor. We set the `alpha` value to 0.95 so that there will be around $10\%$ non-zero elements in the resulting $Q$ matrix.

- The coefficients $c$ in the objective function: $c_j \sim \mathcal{N}(0, 0.1^2)$.

- The non-zero elements in the coefficient matrix: $A_{ij} \sim \mathcal{N}(0, 1)$. The coefficient matrix $A$ contains 100 non-zero elements. The positions are sampled randomly.

- The right hand side $b$ of the linear constraints: $b_i \sim \mathcal{N}(0, 1)$.

- The constraint types $\circ$. We first sample equality constraints following the Bernoulli distribution $Bernoulli(0.3)$. Then other constraints takes the type $\leq$. Note that this is equivalent to sampling $\leq$ and $\geq$ constraints separately with equal probability, because the elements in $A$ and $b$ are sampled from symmetric distributions.

- The lower and upper bounds of variables: $l_j, u_j \sim \mathcal{N}(0, 10^2)$. We swap their values if $l_j > u_j$ after sampling.

- (MI-LCQP only) The variable types are randomly sampled. Each type (*continuous* or *integer*) occurs with equal probability.

After instance generation is done, we collect labels, i.e., the optimal objective function values and optimal solutions, using one of the commercial solvers.

**LCQP instance generation for generalization experiments.** In this setting, we only sample different coefficients $c$ for different LCQP instances. We sample other components only once, i.e., $Q$, $A$, $b$, $l$, $u$ and $\circ$ in (2), and keep them constant and shared by all instances. We also slightly adjust the distributions from which these components are sampled as described below.

- Matrix $Q$. We follow the same sampling scheme as above.

- The coefficients $c$ in the objective function: $c_j \sim \mathcal{N}(0, 1/n)$.

- The non-zero elements in the coefficient matrix: $A_{ij} \sim \mathcal{N}(0, 1/n)$. The coefficient matrix $A$ contains 100 non-zero elements. The positions are sampled randomly.

- The right hand side $b$ of the linear constraints: $b_i \sim \mathcal{N}(0, 1/n)$.

- The constraint types $\circ$. We follow the same sampling scheme as above.

- The lower and upper bounds of variables: $l_j, u_j \sim \mathcal{N}(0, 1)$. We swap their values if $l_j > u_j$ after sampling.

For the generalization experiments, we first generate 25,000 LCQP instances for training, and then take the first 100/500/25,00/5,000/10,000 instances to form the smaller training sets. This ensures that the smaller training sets are subsets of the larger sets. The test set contains 1,000 instances that are generated separately.

**Portfolio optimization formulation and instance generation.** The portfolio optimization problems are formulated as below.

$$\min_{x,y} \quad \frac{1}{2}x^\top D x + \frac{1}{2}y^\top y - \mu^\top x \tag{21}$$
$$\text{s.t.} \quad y = Fx, \quad \mathbf{1}^\top x = 1, \quad x \geq 0$$

Here $x \in \mathbb{R}^s$ and $y \in \mathbb{R}^t$ are the optimization variables, $D \in \mathbb{R}^{s \times s}$ is a diagonal matrix with non-negative diagonal elements, $F \in \mathbb{R}^{t \times s}$ is the factor modeling matrix. We generate portfolio optimization instances following the scheme below.

- We set $s = 50$ and $t = 5$, resulting in LCQP instances with $m = 6$ constraints and $n = 55$ variables.

- The diagonal elements of $D$ are independently sampled from uniform distribution: $D_{ii} \sim U(0, \sqrt{t})$. $D$ is then used to form the matrix $Q = \begin{pmatrix} D & \\ & I_t \end{pmatrix}$.

- The coefficients $\mu$ in the objective function: $\mu_j \sim \mathcal{N}(0, 1)$.

- The non-zero elements in the factor modeling matrix $F$: $F_{ij} \sim \mathcal{N}(0, 1)$. The coefficient matrix $F$ contains 25 non-zero elements. The positions are sampled randomly.

**SVM optimization formulation and instance generation.** The support vector machine optimization problems are formulated as below.

$$\min_{x,t} \quad \frac{1}{2}x^\top x + \lambda \mathbf{1}^\top t \tag{22}$$
$$\text{s.t.} \quad t \geq \operatorname{diag}(y)Dx + \mathbf{1}, \quad t \geq 0$$

Here $x \in \mathbb{R}^s$ and $t \in \mathbb{R}^t$ are the optimization variables, $D \in \mathbb{R}^{t \times t}$ is the data matrix, $y \in \mathbb{R}^t$ is the binary label vector, and $\lambda$ is a hyperparameter which we set to $1/2$. We generate SVM optimization instances following the scheme below.

- We set $s = 5$ and $t = 50$.

- The non-zero elements in the data matrix $D$: $D_{ij} \sim \mathcal{N}(-0.1, 0.1)$ for $i \leq t/2$; $D_{ij} \sim \mathcal{N}(0.1, 0.1)$ otherwise. The coefficient matrix $D$ contains 100 non-zero elements. The positions are sampled randomly.

- The binary label vector $y$: $y_i = -1$ for $i \leq t/2$; $y_i = 1$ otherwise.

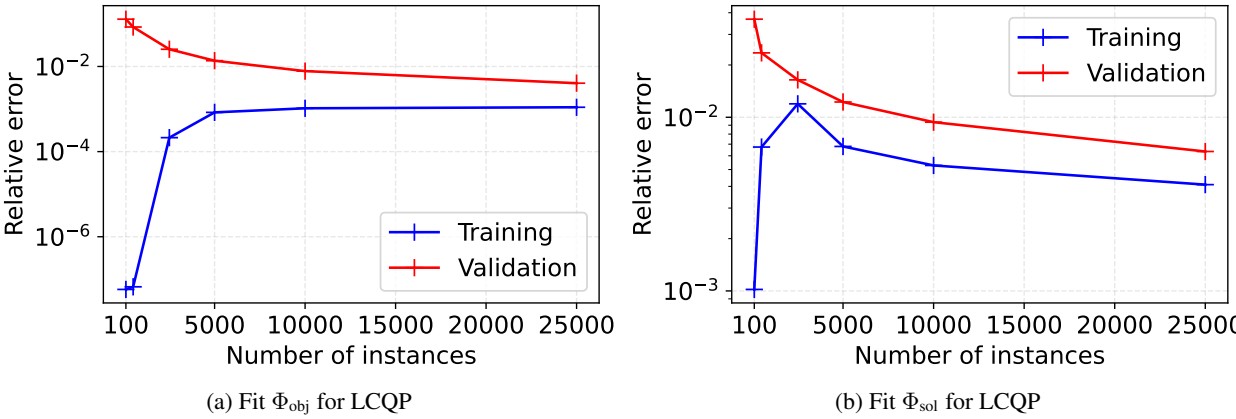

(a) Fit $\Phi_{\text{obj}}$ for LCQP          (b) Fit $\Phi_{\text{sol}}$ for LCQP

*Figure 7.* Training and test errors when training GNNs with an embedding size of 512 on different numbers of LCQP problem instances to fit $\Phi_{\text{obj}}$ and $\Phi_{\text{sol}}$.

### G.2. Details of GNN implementation

We implement GNN with Python 3.9 and TensorFlow 2.16.1 (Abadi et al., 2016). Our implementation is built by extending the GNN implementation in Gasse et al. (2019).[7] The embedding mappings $f_0^V, f_0^W$ are parameterized as linear layers followed by a non-linear activation function; $\{f_l^V, f_l^W, g_l^V, g_l^W, g_l^Q\}_{l=1}^L$ and the output mappings $r_1, r_2$ are parameterized as 2-layer multi-layer perceptrons (MLPs) with respective learnable parameters. The parameters of all linear layers are initialized as orthogonal matrices. We use ReLU as the activation function.

In our experiments, we train GNNs with embedding sizes of 64, 128, 256, 512 and 1,024. We show in Table 3 the number of learnable parameters in the resulting network with each embedding size.

*Table 3.* Number of learnable parameters in GNN with different embedding sizes.

| Embedding size | Number of parameters |
| --- | --- |
| 64 | 112,320 |
| 128 | 445,824 |
| 256 | 1,776,384 |
| 512 | 7,091,712 |
| 1,024 | 30,436,352 |

### G.3. Details of GNN training

We adopt Adam (Kingma & Ba, 2014) to optimize the learnable parameters during training. We use an initial learning rate of $5 \times 10^{-4}$ for all networks. We set the batch size to 2,500 or the size of the training set, whichever is the smaller. In each mini-batch, we combine the graphs into one large graph to accelerate training. All experiments are conducted on a single NVIDIA Tesla V100 GPU.

We use mean squared relative error as the loss function, which is defined as

$$L_{\mathcal{G}}(F_W) = \mathbb{E}_{G \sim \mathcal{G}} \left[ \frac{\|F_W(G) - \Phi(G)\|_2^2}{\max(\|\Phi(G)\|, 1)^2} \right], \tag{23}$$

where $F_W$ is the GNN, $\mathcal{G}$ is a mini-batch sampled from the whole training set, $G$ is a problem instance in the mini-batch $\mathcal{G}$, and $\Phi(G)$ is the label of instance $G$. During training, we monitor the average training error in each epoch. If the training loss does not improve for 50 epochs, we will half the learning rate and reset the parameters of the GNN to those that yield the lowest training error so far. We observe that this helps to stabilize the training process significantly and can also improve the final loss achieved.

---

[7]See https://github.com/ds4dm/learn2branch.

### G.4. Generalization results on LCQP

Figure 7 shows the variations of training and test errors when training GNNs of an embedding size of 512 on different numbers of LCQP problem instances. We observe similar trends for both prediction tasks, that the generalization gap decreases and the generalization ability improves as more instances are used for training. This result implies the potential of applying trained GNNs to solve QP problems that are unseen during training but are sampled from the same distribution, as long as enough training instances are accessible and the instance distribution is specific enough (in contrast to the generic instances used in experiments of Figure 2 and 3).

### G.5. Details for on Maros-Meszaros test set

To show the fitting ability of GNNs on more realistic QP problems, we train GNNs on the Maros and Meszaros Convex Quadratic Programming Test Problem Set (Maros & Mészáros, 1999), which contains 138 quadratic programs that are designed to be challenging. We apply equilibrium scaling to each problem and also scale the objective function so that the $Q$ matrix will not contain too large elements. We collect the optimal solutions and objective values of the test instances using an open-sourced QP solver called PIQP (Schwan et al., 2023), which is benchmarked to achieve best performances on the Maros Meszaros test set among many other solvers (Caron et al., 2024). PIQP solves 136 problem instances successfully, which are then used to train four GNNs with embedding size of $64, 128, 256, 512$. The training protocol follows the experiments using synthesized QP instances in Section 5.

