# OpenReview forum: "Expressive Power of Graph Neural Networks for (Mixed-Integer) Quadratic Programs"
_ICML.cc/2025/Conference — ICML 2025 poster_

### Official Review · Reviewer_4Q8A · 2025-02-26

**Overall Recommendation:** 4

**Summary:**

In this work the authors study quadratic optimization problems with linear constraints and answer the question if the objective value, feasibility and optimal solution of this problem class can be approximated by a class of graph neural networks. More precisely they study the standard graph representation of LCQPs and show that if all decision variables are continuous there exists graph neural networks which (i) can approximate the objective value up to an arbitrary accuracy with high probability, (ii) can decide if the corresponding problem is feasible with high probability, and (iii) can predict the optimal solution with minimal l2 norm up to an arbitrary accuracy with high probability. On the contrary the authors show that, if some of the variables may be restricted to be integer, the latter results do not hold in general. However, they can derive subclasses of LCQPs for which (i)-(iii) hold. Finally, the authors showcase approximation capabilities of GNNs in some numerical experiments.

"## update after rebuttal
No update.

**Claims And Evidence:**

All the claims made in the submission are thoroughly explained and proved. The authors even provide experimental results which are in my opinion not necessary for such a theoretical work, since finding the GNN which approximates the metrics of the problem class depends a lot on the learning algorithm and hyperparameter setting and is a task for a completely different study.

**Essential References Not Discussed:**

No references.

**Experimental Designs Or Analyses:**

Soundness of the experimental design is very satisfying, see comments above.

**Methods And Evaluation Criteria:**

The work is mainly theoretical and all results are properly explained and proved. The experimental confirmation of the expressiveness of the GNNs is a nice addition to the theoretical results showing that even standard training methods already lead to GNNs which nearly perfect approximate the desired property of the LCQPs. However, in practical settings the more important question is if the GNNs generalize well on unseen data, i.e., did the learning algorithm indeed find a GNN which has the theoretically desired expressiveness on the whole problem class. The latter question is not answered in detail and is in my opinion not needed for this work.

**Other Comments Or Suggestions:**

- Line 47: "with with"
- Line 64: exploits -> exploit

**Other Strengths And Weaknesses:**

Strengths:
- the paper is very well written and all concepts and proofs are mathematically thoroughly presented. I could not find any mathematical flaws in the main paper.

**Questions For Authors:**

- What was the main difficulty to extend the work of Chen et al. (2023) to the quadratic setting?

**Relation To Broader Scientific Literature:**

The work closely builds on the work of Chen et al (2023) where the same methodology is used for linear problems. However, extending the work to quadratic problems is valuable and relevant for the research field.

**Theoretical Claims:**

I briefly checked the proofs in the Appendix without checking every detail.

---

> ### Author Rebuttal · Authors · 2025-04-01
>
> __Reply to "Methods And Evaluation Criteria":__ Thank you for the encouraging comments, and we really appreciate it. We agree that generalization analysis is an important direction, especially for structured problems like LCQPs and MI-LCQPs, and we will highlight it as a key avenue for future work in the revised paper.
>
> While we do not provide a theoretical generalization analysis, we included an empirical study in Section F.4 and Figure 6 (Appendix F), to investigate the generalization performance of GNNs on QP tasks. The results show that as the number of training instances increases, the generalization gap decreases and performance improves. This suggests that GNNs have the potential to generalize to unseen QPs drawn from the same distribution, provided sufficient and appropriately sampled training data.
>
> __Reply to "Other Comments Or Suggestions":__ Thanks for the detailed comments, and we will fix these typos following your suggestions.
>
> __Reply to "Questions For Authors":__ We acknowledge that our proofs use a similar high-level framework to those in Chen et al. (2023a;b). However, we would like to respectfully emphasize that there are several important and non-trivial differences in the technical details. These include:
> * We have richer counterexamples for MI-LCQPs (Chen et al. (2023b) only have counterexamples for the feasibility).
> * We establish the connection between the expressive power of the WL test and the specific properties of LCQPs and MI-LCQPs. In particular, we have some new analysis handling the quadratic terms in Appendix A that is not directly from Chen et al. (2023a;b), and we show that the GNN approximation theory for feasibility/optimal objective can hold with a weaker assumption than the optimal solution (GNN-analyzability vs GNN-solvability), which is also not covered in Chen et al. (2023b).
> * We design the graph representation (sometimes with hyperedges, see Appendix E) to fully encode all elements of these quadratic programs; and we develop theoretical analysis involving hyperedges for QCQPs. Technically, the representation and analysis of hyperedges in Appendix E are significantly different.

---

### Official Review · Reviewer_tWyf · 2025-03-07

**Overall Recommendation:** 3

**Summary:**

This work proves that message-passing GNNs can universally represent fundamental properties of quadratic programs, including feasibility, optimal objective values, and optimal solutions. They also prove that GNNs are not universal for mixed-integer problems.

**Claims And Evidence:**

Yes.

**Essential References Not Discussed:**

This work also provides NN/GNN universality and complexity for solving convex quadratic programs [1].

[1] Yang, L., Li, B., Ding, T., Wu, J., Wang, A., Wang, Y., ... & Luo, X. (2024). An Efficient Unsupervised Framework for Convex Quadratic Programs via Deep Unrolling. arXiv preprint arXiv:2412.01051.

**Experimental Designs Or Analyses:**

Yes.

**Methods And Evaluation Criteria:**

Yes.

**Other Comments Or Suggestions:**

N/A

**Other Strengths And Weaknesses:**

Strengths:
1. The paper is well-written.
2. The theoretical results on GNN universality for QP are solid and important for its applications.

Weakness:
1. The authors mention a concurrent work by Wu et al. (2024) that explores a tripartite graph representation and its associated GNN for QCQP, investigating its expressive power (line 105). Since convex QP is a special case of QCQP, the authors should compare and discuss their results and methodology with those of Wu et al. (2024).
2. Beyond establishing universality, the parameter complexity of the GNN is not addressed. Understanding this complexity is crucial for evaluating its superiority compared to standard neural networks, which are also universal approximators for continuous mappings.
3. Although the framework is demonstrated for LP and QP, it is unclear whether this graph representation approach can be generalized to more complex problems, such as polynomial optimization. Clarification or extension to these cases would strengthen the paper's contribution.

**Questions For Authors:**

1. Are there any real-world MI-LCQPs that are GNN-friendly?
2. Is there any theoretical advantage to using GNNs instead of regular neural networks when solving QPs?
3. Could the authors clarify the specific methodological contributions of this work compared with the established methods in [1,2]?

[1] Chen, Z., Liu, J., Wang, X., Lu, J., and Yin, W. On representing linear programs by graph neural networks. In The Eleventh International Conference on Learning Representations, 2023a.
[2] Chen, Z., Liu, J., Wang, X., Lu, J., and Yin, W. On representing mixed-integer linear programs by graph neural networks. In The Eleventh International Conference on Learning Representations, 2023b.

**Relation To Broader Scientific Literature:**

N/A

**Theoretical Claims:**

No.

---

> ### Author Rebuttal · Authors · 2025-04-01
>
> __To reviewer:__ Thank you for your insightful comments. Due to the 5000-character limit, our responses must be brief, but we’d be happy to elaborate on any specific points in the next stage of rebuttal.
>
> __Reply to "Essential References Not Discussed":__ Thanks for highlighting this relevant work. We’ll cite and discuss it in the revision. This concurrent study [1] proposes an unrolling-based framework for convex LCQPs with explicit complexity bounds, while our work analyzes GNNs’ expressive power for broader problem classes (including mixed-integer LCQPs) with universal approximation results, though without complexity bounds. The two are complementary.
>
> __Reply to Weakness 1:__ We will include a more detailed comparison with Wu et al. (2024) in the revision.
>
> * Methodologies differ: Wu et al. use a tripartite graph for general QCQPs, while we use a variant of bipartite graph tailored to (MI-)LCQPs.
> * Results differ with partial overlap: both cover convex LCQPs, but Wu et al. include quadratic constraints, whereas we handle mixed-integer cases and offer an alternative quadratic constraint approach.
>
> Notably, while Appendix E discusses QCQPs, our method differs structurally: Wu et al. add nodes for quadratic terms, while we use hyperedges, leading to a hypergraph GNN.
>
> __Reply to Weakness 2:__ We agree that analyzing GNN parameter complexity is important. While our current results are non-parametric, we will add a discussion: The algorithm-unrolling paper by Yang et al. [1] provides a way to estimate parameter complexity, as each layer's parameters in [1] are explicitly defined. Moreover, prior work (e.g., [2]) shows that unrolling can be viewed as a structured GNN. These links suggest that one may derive GNN complexity bounds for (MI-)LCQPs from unrolling results.
>
> [2] Li et al. "On the Power of Small-size Graph Neural Networks for Linear Programming." NeurIPS 2024.
>
> __Reply to Weakness 3:__ We'll clarify this in the revision: The graph representation can indeed be extended to polynomial optimization. We suggest modeling terms like $F_{j_1,...,j_k}x_{j_1}\cdots x_{j_k}$ as hyperedges:
>
> - $(w_{j_1},...,w_{j_k})$ for objectives
> - $(v_i,w_{j_1},...,w_{j_k})$ for constraints
>
> Our QCQP analysis (Appendix E) shows a concrete example of this idea: hypergraph GNNs can approximate such properties, suggesting potential for broader extension to convex polynomial optimization.
>
> __Reply to Question 1:__ Yes, there are. In QBLIB, 77 out of 96 binary-variable linear constraint QPs are GNN-friendly. Detailed results can be found in the paragraph “Frequency of GNN-friendly instances in practice” and in Table 2 (Page 22). We'll add this statistic to Section 4.3 (Page 7) in the revision.
>
> __Reply to Question 2:__ Compared to standard NNs, GNNs provide some unique theoretical advantages:
>
> * __Permutation Invariance/Equivariance__
>    - In QPs, swapping variable/constraint order should permute solutions accordingly.
>    - __Standard neural networks lack built-in permutation invariance/equivariance__, requiring explicit training on all $O(n!)$ input permutations. __In contrast, GNNs inherently maintain this property__ through their graph-based architecture, automatically handling variable/constraint reordering without additional training.
>
> * __Scalability to Varying Sizes__
>    - In GNNs, the learnable functions ($g$'s and $r$'s) and their parameters are shared across nodes and do not depend on the specific index $i,j$, enabling **direct application to new problem sizes** without retraining.
>    - Standard NNs need fixed input/output dimensions, requiring architectural changes for size variations.
>
> These points were briefly mentioned at Lines 64–67 (left column), and we will expand them in the revision.
>
> __Reply to Question 3:__ While our proofs share a high-level framework with Chen et al. (2023a; b), we highlight several non-trivial technical advances:
>
> 1. **Enhanced Counter-Examples**
>    - Chen et al. (2023b) only show feasibility counterexamples
>    - We provide richer counterexamples (feasibility, optimal objective, and optimal solution) for MI-LCQPs
>
> 2. **New Theoretical Connections**
>    - Connect the expressive power of the WL test to key properties of (MI-)LCQPs.
>    - Novel analysis of quadratic terms (Appendix A)
>    - Prove GNN approximation under weaker assumptions (GNN-analyzability vs GNN-solvability). Chen et al. (2023b) always assume GNN-solvability.
>
> 3. **Hypergraph Innovations**
>    - Develop novel hyperedge representations and analysis for QCQPs (Appendix E)
>    - Significant technical differences from Chen et al. (2023a; b) that do not involve hypergraphs.
>
> While our work builds on prior studies, we believe it makes valuable contributions. Given the importance of QPs, a theoretical foundation for GNNs in this setting is both timely and impactful. Our proposed criteria—GNN-analyzable and GNN-solvable—serve as practical tools for assessing datasets and diagnosing issues in MI-LCQP training.

---

### Official Review · Reviewer_PHmc · 2025-03-12

**Overall Recommendation:** 2

**Summary:**

This paper provides a theoretical analysis to investigate the expressive power of standard Message-Passing GNNs (MPGNNs) in solving the Linearly Constrained Quadratic Program (LCQP) and Mixed-Integer (MI) LCQP tasks. Specifically, the paper focuses on three mappings
with MPGNNs: feasibility mapping, optimal objective mapping, and optimal solution mapping. The theoretical findings are well-validated with a comprehensive experiment.

**Claims And Evidence:**

Yes. The claims are supported by clear and convincing evidence.

**Essential References Not Discussed:**

None witnessed

**Experimental Designs Or Analyses:**

Yes. The paper conducts neumerical experiments to validate the GNN's expressive power to fit optimal objective mapping and optimal solution mapping on the LCQP amd MI-LCQP. The authors utilize both real-world datasets and synthetic dataset. For the experiment results of real-world datasets (Appendix F.5), in general, the training errors are high. This may indicate that standard MPGNNs may struggle with real-world QPs. It would be better to provide a discussion to analyze why.

**Methods And Evaluation Criteria:**

Yes.

**Other Comments Or Suggestions:**

The writings could be improved, e.g., typos. (e.g. Line 668 “Fixe”) and having legends for figures.

**Other Strengths And Weaknesses:**

Strengths:

1)Though this work mainly focuses on the theoretical part, the identification of "GNN-friendly" subclasses (Definition 4.4) and the criteria for verifying them (Section 4.2) provide insights into real-world tasks;

2) The numerical validation that was conducted on both synthetic and benchmark datasets (Maros-Meszaros) strengthens the practical effect.


Weakness: In the preliminaries section, Q is defined to be symmetric, however, it’s defined to be positive semidefinite in Theorem 3.3 and 3.4. It’s confusing whether “(Q is positive semidefinite almost surely)” in Theorem 3.3 refers to a real-world fact or the previous condition.
If it’s the latter, then the theorems do not guarantee the affirmative answer when Q is symmetric but not positive semidefinite. If it’s the former, the fact should be stated clearly out of the theorem.

**Questions For Authors:**

The paper proves the expressive power limitations of standard MPGNNs for LCQP problems through WL equivalence (Theorem 3.2–3.4). Recent works have shown that high-order or hypergraph GNNs offer greater expressive power than standard MPGNNs[1]. Is it possible to
apply the approach used in the paper to these GNNs to address such limitations? In addition, for Quadratically Constrained Quadratic Programs (QCQP) discussed in Appendix E, constraints may involve interactions between multiple variables (e.g., xixj <= b), and hyperedges in a hypergraph can naturally represent such relationships. It would be interesting to investigate the possibility of providing a universal analysis for LCQP and QCQP.

[1] Feng, Jiarui, et al. "How powerful are k-hop message passing graph neural networks." Advances in Neural Information Processing Systems 35 (2022): 4776-4790

**Relation To Broader Scientific Literature:**

The paper explore the MPGNNs' expressive power for LP and MILP to the LCQP tasks. Specifically, the paper utilizes the Weisfeiler-Lehman (WL) test to prove that separation power (Section 3, Appendix A) for non-linear QP. In addition, the differences
between LCQP and MI-LCQP universality (Theorems 3.2–3.4 vs. Propositions 4.1–4.3) are well-explained, particularly the counterexamples that demonstrate MPGNNs limitations in mixed-integer settings.

However, Theorems and their proofs in sections 3 and 4 for QP and MIQP are almost direct extensions from (Chen et al. 2023a;b) which focused on LP and MILP, making this work incremental to previous works and thus may lack novelty and non-trivial contributions.

In addition, high expressiveness of the hypothesis space does not necessarily lead to better generalization performance in theory (universal neural models are in fact easy to acquire). It would be a great plus to complete the theoretical analysis by providing generalization bounds.

**Theoretical Claims:**

Yes. I have checked the proofs listed in section 3 and section 4.

---

> ### Author Rebuttal · Authors · 2025-04-01
>
> __To reviewer:__ Thank you for your valuable comments. Due to the 5000-character limit, our responses must be brief, but we’d be happy to elaborate on any specific points in the next stage of rebuttal.
>
> __Reply to "Experimental Designs Or Analyses":__ We agree that the training error on the Maros-Mészáros test set is relatively higher compared to synthetic datasets. This is primarily due to its inherent diversity. The 138 quadratic programs are sourced from multiple domains (CUTE library, Brunel Optimization Group, and seven other institutions), making the dataset highly **heterogeneous** with few instances per application scenario.
>
> This contrasts with prior successful empirical work on learning to solve QPs (Nowak et al., 2017; Wang et al., 2020b, 2021; Qu et al., 2021; Gao et al., 2021; Tan et al., 2024), which typically focus on single domains with more consistent problem structures.
>
> While we recommend practitioners train GNNs on domain-specific instances, our theoretical goal requires validating GNNs' **general** ability to represent (MI-)LCQPs. Despite the dataset's challenges, we observe consistent improvement in GNN expressivity with increased model capacity (e.g., larger embeddings), mirroring trends in synthetic experiments.
>
> Finally, we note the issue of **numerical instability**. Unlike synthetic datasets, Maros-Mészáros problems involve coefficients with wide-ranging magnitudes (e.g., from -780600000 to 1, after being converted to one-sided), making training numerically challenging. Despite this, GNNs demonstrate the ability to fit optimal objective values and solutions.
>
> __Reply to "Relation To Broader Scientific Literature":__ While our proofs share a high-level framework with Chen et al. (2023a; b), we highlight several non-trivial technical advances:
>
> 1. **Enhanced Counter-Examples**
>    - Chen et al. (2023b) only show feasibility counterexamples
>    - We provide richer counterexamples (feasibility, optimal objective, and optimal solution) for MI-LCQPs
>
> 2. **New Theoretical Connections**
>    - Connect the expressive power of the WL test to key properties of (MI-)LCQPs
>    - Novel analysis of quadratic terms (Appendix A)
>    - Prove GNN approximation under weaker assumptions (GNN-analyzability vs GNN-solvability). Chen et al. (2023b) always assume GNN-solvability.
>
> 3. **Hypergraph Innovations**
>    - Develop novel hyperedge representations and analysis for QCQPs (Appendix E)
>    - Significant technical differences from Chen et al. (2023a; b) that do not involve hypergraphs.
>
> While our work builds on prior studies, we believe it makes valuable contributions. Given the importance of QPs, a theoretical foundation for GNNs in this setting is both timely and impactful. Our proposed criteria—GNN-analyzable/solvable—serve as practical tools for assessing datasets and diagnosing issues in MI-LCQP training.
>
> Additionally, we agree that generalization analysis is crucial, and will highlight this as key future work in the revision.
>
> __Reply to Weakness:__ It is the latter. We will further clarify the convex assumption in the revised manuscript:
> 1. **Page 2 (Contributions):** Clarify convexity assumption and nonconvex counterexample
> 2. **Page 4:** Rename Section 3 → "Universal approximation for **convex** LCQPs"
> 3. **Page 5:** Add nonconvex LCQP counterexample showing GNN indistinguishability despite different optima.
>
> Consider a convex LCQP
> $$\min~~ \frac{1}{2} \begin{bmatrix}x_1 & x_2\end{bmatrix} \begin{bmatrix}1 & 0 \\\\ 0 & 1\end{bmatrix} \begin{bmatrix}x_1 \\\\ x_2\end{bmatrix}, \quad\text{s.t.}~~ -1\leq x_1,x_2\leq 1,$$
> and a nonconvex LCQP
> $$\min~~ \frac{1}{2} \begin{bmatrix}x_1 & x_2\end{bmatrix} \begin{bmatrix}0 & 1 \\\\ 1 & 0\end{bmatrix} \begin{bmatrix}x_1 \\\\ x_2\end{bmatrix}, \quad\text{s.t.}~~ -1\leq x_1,x_2\leq 1.$$
> These two LCQPs are indistinguishable by GNNs, and they have different optimal objective/solution.
>
> __Reply to "Other Comments Or Suggestions":__ We'll fix these typos in the revision.
>
> __Reply to "Questions For Authors:"__ Thank you for this insightful comment. We fully agree that $k$-hop GNNs exhibit stronger separation power than MP-GNNs. In the proof of Prop. 4.2 (Appendix B), we construct MI-LCQP instances with distinct optimal objectives that MP-GNNs cannot distinguish. But **3-hop GNNs can** distinguish them. Crucially, when we scale these examples to 10 variables/10 constraints (one graph with 20 connected nodes vs. two 10-node components), **3-hop GNNs fail** while **5-hop GNNs succeed**, confirming that larger $k$ reduces unsolvable cases. We will include this analysis in the revision.
>
> For QCQPs, Appendix E.1 presents a hypergraph GNN framework, with universal approximation analysis (Theorems E.2–E.4). While (MI-)LCQPs anchor our main narrative, we agree that unifying LCQP/QCQP/higher-order extensions via hypergraph GNNs is a compelling direction for future work, which we will emphasize.

---

### Official Review · Reviewer_ADeR · 2025-03-13

**Overall Recommendation:** 4

**Summary:**

The paper establishes that message-passing GNNs can express the feasibility, optimal value and optimal solution of convex linearly constraint quadratic programs as well as of mixed-integer linearly constraint quadratic programs, if they adhere to certain conditions often true in practice. Negative results are provided for mixed-integer linearly constraints quadratic programs in general. Numerical experiments validate the approximation results.

**Claims And Evidence:**

In the introduction, it is claimed that GNNs can accurately predict the feasibility, optimal value and solution of a linearly constraint QP. However, the last two claims require the additional assumption of a convex quadratic program (i.e., positive semidefinite $Q$). This should be reflected in the claim. Other than this, all claims are clear and provided with convincing evidence.

**Essential References Not Discussed:**

All essential related work has been discussed.

**Experimental Designs Or Analyses:**

Yes, some issues:

**(I)** I'm not convinced of omitting the experiments where GNNs are used to fit $\Phi_{feas}$ based on arguing that feasibility falls to the case of LP and MILP (Chen et al. 2023). This is as due to the fact that the input graph representing a LCQP / MI-LCQP is still different to the one for an LP / MILP and thus, the empirical behavior of a GNN on the changed input graph is still interesting.

**(II)** Fig 2 to 4 and Tabl. 1 are missing standard deviations.

**(III)** The experimental details mention that four GNNs are trained and averaged over all instances during training. I would have assumed that the same experiments are also repeated for several seeds, which seems not to be the case.

**(IV)** While for showing approximation results, it is not necessary for the GNNs to be applied to unseen instances, I would be curious about how well the GNNs actually generalize to unseen instances. This is slightly touched upon in F.4 where results on the validation set are shown, but no results on a held-out test set are provided. It would be easy to generate a few more random instances to test this behavior.

**(V)** GNNs expressiveness results are not tested on practical MI-LCQP instances. Have you considered GNN-solvable/analyzable instances from QPLib?

**Methods And Evaluation Criteria:**

Yes, for details see Experimental Designs Or Analyses.

**Other Comments Or Suggestions:**

* The definition of $\Phi_{sol}$ for an MI-LCQP is quite hidden in the mids of App. C, it would be helpful to make it more prominent.
* In Section 2, the mappings $g$ and $r$ are used before they are introduced.
* Define the $\succeq$ operator in Line 154 (second column). Here it is used for a positive semi-definite matrix. However, in the ML literature, it is sometimes used for matrix inequalities.
* Line 047 "with with"
* Section 3 header misses a "i" in "Universal"
* Line 061 and 062, use \citet for Wang & Yu's works
* Line 169/170 "there always be" -> "there always is"

**Other Strengths And Weaknesses:**

The paper is written in a very clear fashion and good to follow. As the results on LCQPs and MI-LCQPs have not been known, the results are significant. The idea to the research question may not be the most original, as after this problem has been approached for LPs and MILPs by Chen et al. (2023a, b) it is natural to extend the question to other optimization problems. However, it required new ideas to proof and is still an important and significant result, providing important theoretical grounding and guiding for the empirical developments in the field.

**Questions For Authors:**

**(I)** It seems that predicting the optimal objective value / solution is more difficult than predicting if an LCQP is feasible (for the first, the requirement of convexity is needed, for the second not). Can you comment on how the three problems relate to one another in some sense of difficulty hierarchy?

**(II)** Do you have a negative result for non-convex LCQPs?

**(III)** In Lines 188-192 (second column) the authors state that any two LCQP-graphs that are indistinguishable by the WL test, or equivalently by all GNNs, must have identical optimal value and solution. As Xu et al. (2019) showed that message-passing GNNs are upper bounded by the 1-WL but not necessarily reach its expressivity, does the proposed family of GNN architectures reach 1-WL expressivity?

**(IV)** Section 2, node-level output: There is some issue with the definition of $y$. It is indexed by $j$ even though defined for each $i \in V$. Shouldn't it be defined for each $j \in W$?

**(V)** Can you give some intuition on what "GNN-analyzable" means?

**Relation To Broader Scientific Literature:**

As universal approximation of message-passing GNNs has been investigated for LPs and MILPs, extending the study to QPs seems natural and important, especially as there are several works, also cited in this work, on empirically learning to solve QPs using GNNs.

**Theoretical Claims:**

I checked the argumentations in the main draft, I did go over some arguments in the appendix, but did not check the proofs in the appendix in detail.

Two minor issues:

**(I)** Lines 357-360 (left column) reads as if in the GNN-solvable case, GNNs can approximate the optimal solution, but not feasibility. Given Prop. D.1, this should not be true. I would make this more clear.

**(II)** I think the proof leading to the statement that GNN-solvable and GNN-analyzable instances make up the majority of the MI-LCQP set could be assumed very synthetic. Coefficients in the objective would never be set to irrational numbers, it is rather reasonable to even assume that in practice $c$ may be the same for many $j$. The empirical investigation on QPLIB is interesting and supports that the assumption of randomly sampling $c$ from $\mathbb{R}^n$ and using properties of $\mathbb{R}$ seems too strong to explain the real-world occurrences of GNN-solvable/analyzable instances.

---

> ### Author Rebuttal · Authors · 2025-04-01
>
> __To reviewer:__ Thank you for your detailed comments! Due to the 5000-character limit, our responses must be brief, but we’d be happy to elaborate further in the next rebuttal stage.
>
> __Reply to "Claims And Evidence":__ We will clearly state the convex assumption in the revision:
> 1. **Page 2 (Contributions):** Clarify convexity assumption and nonconvex counterexample
> 2. **Page 4:** Rename Section 3 → "Universal approximation for *convex* LCQPs"
> 3. **Page 5:** Add nonconvex LCQP counterexample showing GNN indistinguishability despite different optima.
>
> Consider a convex LCQP
> $$\min~~ \frac{1}{2} \begin{bmatrix}x_1 & x_2\end{bmatrix} \begin{bmatrix}1 & 0 \\\\ 0 & 1\end{bmatrix} \begin{bmatrix}x_1 \\\\ x_2\end{bmatrix}, \quad\text{s.t.}~~ -1\leq x_1,x_2\leq 1,$$
> and a nonconvex LCQP
> $$\min~~ \frac{1}{2} \begin{bmatrix}x_1 & x_2\end{bmatrix} \begin{bmatrix}0 & 1 \\\\ 1 & 0\end{bmatrix} \begin{bmatrix}x_1 \\\\ x_2\end{bmatrix}, \quad\text{s.t.}~~ -1\leq x_1,x_2\leq 1.$$
> These two LCQPs are indistinguishable by GNNs, and they have different optimal objective/solution.
>
> __Reply to "Theoretical Claims":__
>
> __(I)__ We will clarify in our revision that in the GNN-solvable case, GNNs can approximate *not only* the optimal solution but also the feasibility and the optimal objective, as GNN solvability implies GNN-analyzability (Prop. D.1).
>
> __(II)__ We acknowledge this conclusion is more synthetic, hence its appendix placement. In Section 4.3, we'll clarify:
> 1) GNN-solvable/analyzable cases dominate *under specific distributional assumptions*
> 2) *Real-world* datasets (e.g., QPlib) may contain GNN-insolvable cases (Appendix D)
>
> __Reply to "Experimental Designs Or Analyses":__
>
> __(I)__ While (MI)LP and (MI-)LCQP input graphs differ due to $Q$, our GNN architecture can emulate LP/MILP cases by setting $g_l^Q = 0$, making the architectures equivalent when $Q$ is ignored.
>
> __(II)__ We re-measure the solving times in Table 1 and present the average solving times and standard deviations below. While the solving times are different due to changes in hardware environment and system load, the advantage of GNN with large batch sizes is consistent. We will update Table 1 and Fig. 2-4.
>
> | GNN | BS=1 | BS=10 | BS=100 | BS=1000 | OSQP |
> |---|---|---|---|---|---|
> | Time | 53.62±16.72 | 5.37±1.87 | 0.504±0.142 | 0.089±0.002 | 4.48±3.62 |
>
> __(III)__ We run two more sets of experiments on different random seeds, which decide the problem generation, GNN initialization, and stochastic optimization. The results are consistent. For example, over three experiments of training a GNN with an embedding size of 256 to fit the optimal solutions of 500 LCQP problems, the average relative errors we can achieve are $2.83\times 10^{-3}$, $2.69\times 10^{-3}$ and $2.84\times 10^{-3}$. We will add the full results in the revision.
>
> __(IV)__ We apologize for using the confusing term "validation set" in F.4. The set was never seen during training. Hence, the results in Fig. 4 are the generalization results that you asked for. We will change the terms to avoid confusion.
>
> __(V)__ We performed GNN training on 73 GNN-solvable instances from QPLib that provide the optimal solutions and objectives. We train GNNs of embedding sizes 128, 256, and 512 to fit the objectives and solutions. The training errors we achieved are shown below. GNNs can fit the objective values well and demonstrate the ability to fit solutions. The results show the model capacity improves as the model size increases. We will include the training curves in the revision.
>
> | emb_size | 128 | 256 | 512 |
> |---|---|---|---|
> | objective | 3.28E-04 | 9.05E-07 | 4.00E-07 |
> | solution | 6.57E-01 | 6.72E-01 | 6.06E-01 |
>
> __Reply to "Other Comments Or Suggestions":__ We will revise accordingly.
>
> __Reply to "Questions For Authors":__
>
> __(I)__ The difficulty hierarchy differs by problem type:
>
> - **LCQP:** feas (Assump 3.1) < obj (+convexity) < sol (+feasible/bounded)
>
> - **MI-LCQP:** feas/obj (Assump 4.5 + GNN-analyzable) < sol (+GNN-solvable)
>
> Convexity isn't required for MI-LCQP due to inherent integer non-convexity.
>
> __(II)__ Yes, it is presented at the beginning of our rebuttal.
>
> __(III)__ *Theoretically*, our GNN architecture (Page 3) achieves 1-WL expressivity when mappings ($g$'s, $r$'s) are injective (Xu et al., 2019), satisfied by our continuous function assumption.
>
> *Practically*, MLP implementations limit expressivity, but with sufficiently large MLPs, the expressivity can closely approximate that of 1-WL on specific datasets, as Section 5 shows.
>
> __(IV)__ We will correct it.
>
> __(V)__ *GNN-solvability* requires all variable nodes to be distinguishable (no symmetry), while *GNN-analyzability* permits some symmetry, and requires that edges with identical node-color pairs must share weights. (For example, if two edges both connect a blue node to a red node, they must have the same weight) This requirement implies WL-equivalent nodes share edge-level properties (see Fig.5).

---

> > ### Comment · Reviewer_ADeR · 2025-04-02
> >
> > I thank the authors for the rebuttal that addressed my questions and concerns. Given the answers will be reflected in the camera-ready version, I'm happy to increase my score.

---

> > > ### Author Response · Authors · 2025-04-02
> > >
> > > Dear Reviewer,
> > >
> > > Thank you for your positive feedback and for increasing the score. We’re glad our responses addressed your concerns and will ensure all changes are reflected in the final version. We appreciate your time and support.
> > >
> > > Best,
> > >
> > > Authors

---

### Decision · Program_Chairs · 2025-05-01

**Decision:**

Accept (poster)

**Comment:**

This paper investigates the theoretical expressiveness of message-passing Graph Neural Networks (GNNs) in approximating key properties of linearly constrained quadratic programs (LCQPs) and their mixed-integer counterparts (MI-LCQPs). The authors demonstrate that GNNs can approximate feasibility, optimal objective values, and optimal solutions in the convex LCQP setting. For the more challenging MI-LCQP case, they introduce novel concepts of GNN-analyzability and GNN-solvability, identifying subclasses where GNNs retain expressive power. Theoretical insights are backed by experimental validations on synthetic and real-world datasets.

Overall the reviewers liked the paper (with one weak reject though) and I tend to agree with slight reservations here and there. The discussion was good. All things considered I will recommend weak acceptance.